# EMS1 and BRI1 control separate biological processes via extracellular domain diversity and intracellular domain conservation

Bowen Zheng[1,2], Qunwei Bai [1,2], Lei Wu[1], Huan Liu[1], Yuping Liu[1], Weijun Xu[1], Guishuang Li[1], Hongyan Ren[1], Xiaoping She[1] & Guang Wu [1]

In flowering plants, EMS1 (Excess Microsporocytes 1) perceives TPD1 (Tapetum Determinant 1) to specify tapeta, the last somatic cell layer nurturing pollen development. However, the signaling components downstream of EMS1 are relatively unknown. Here, we use a molecular complementation approach to investigate the downstream components in EMS1 signaling. We show that the EMS1 intracellular domain is functionally interchangeable with that of the brassinosteroid receptor BRI1 (Brassinosteroid Insensitive 1). Furthermore, expressing *EMS1* together with *TPD1* in the *BRI1* expression domain could partially rescue *bri1* phenotypes, and led to the dephosphorylation of BES1, a hallmark of active BRI1 signaling. Conversely, expressing *BRI1* in the *EMS1* expression domain could partially rescue *ems1* phenotypes. We further show that *PpEMS1* and *PpTPD1* from the early land plant *Physcomitrella patens* could completely rescue *ems1* and *tpd1* phenotypes, respectively. We propose that EMS1 and BRI1 have evolved distinct extracellular domains to control different biological processes but can act via a common intracellular signaling pathway.

[1] College of Life Sciences, Shaanxi Normal University, 710119 Xi'an, Shaanxi Province, China. [2]These authors contributed equally: Bowen Zheng, Qunwei Bai. Correspondence and requests for materials should be addressed to X.S. (email: shexiaoping@snnu.edu.cn) or to G.W. (email: gwu3@snnu.edu.cn)

In angiosperms, the development of male reproductive organs is precisely controlled to achieve successful fertilization and reproduction[1–4]. The tapetum, which is required for pollen production, provides pollen nutritive support, remodels the callose coat surrounding microsporocytes and tetrads, and synthesizes most components of the pollen wall[2,5–9]. Anthers in *ems1* (*Excess Microsporocytes 1*), also known as *exs* (*Extra Sporogenous Cells*), and *tpd1 (Tapetum Determinant 1)* mutants have no tapetal cells; instead, they produce excess microsporocytes[6,10,11]. TPD1 is secreted from microsporocyte precursors and then activates EMS1, which is localized at the plasma membrane of tapetal precursor cells/tapetal cells[7,12]. The EMS1-TPD1 signaling pathway initially promotes periclinal division of parietal cells to form tapetal precursor cells, and later determines and maintains the fate of functional tapetal cells[7,12]. The SERK1/2 (Somatic Embryogenesis Receptor-Like Kinase 1,2) LRR-RLKs (Leucine Rich Repeat Receptor-Like Kinases), act as potential co-receptors of EMS1[13].

Given the significance of EMS1 in male fertility, identifying its downstream signaling components is critical. However, since no homozygous seeds can be obtained from *ems1* null mutants, isolation of its downstream components via genetic screens is complicated[10,11]. Furthermore, as *EMS1* expression is tapetum-specific and *ems1* mutants have no tapeta, molecular isolation of downstream components is difficult. Moreover, yeast two-hybrid screen can only isolate direct interactors and affinity purification may generate non-specificity[14–16]. Thus, it was until last year that a putative substrate of EMS1, a family of β-carbonic anhydrases (βCAs), was identified[17]. However, the way βCAs transmit the EMS1 signal to downstream targets is completely unknown[17]. Therefore, identifying additional EMS1-TPD1 signaling components is a necessary but challenging task.

EMS1 belongs to the LRR-X (Leucine Rich Repeats-X) subfamily of receptor-like kinases (RLKs), the largest family of cell surface receptors in land plants[18]. The LRR-RLK-X subfamily also includes BRI1 (Brassinosteroid Insensitive 1) and PSKR1 (Phytosulfokine Receptor 1). Thus, EMS1, BRI1 and PSKR1 have high sequence similarity[18–21]. However, these receptors have distinct biological functions[10,11,20,21]. Current knowledge suggests that RLKs use their versatile extracellular domains (ECDs) to perceive a variety of ligands, which activate conserved intracellular kinase domains (ICDs) to regulate different downstream targets and control distinct biological processes[18,19]. Thus, with greater divergence of the ECDs relative to the ICDs[18,22], it is possible that the ECDs could bind different ligands, while their ICDs still target the same downstream components. This together with differential gene expression controls diverse biological functions[18]. It has been demonstrated that the ICD of BRI1 can be activated in chimeric receptors with the ECDs of distinct RLKs that perceive non-BR ligands or that are coreceptor kinases[22,23]. This finding indicates that the same ECD can activate different ICDs while different ECDs can also activate the same ICD, providing a technical framework to functionally study ECDs and ICDs of a variety of RLKs using chimeric receptors.

Among all the RLKs, BRI1 is one of the best studied receptors. Brassinosteroids (BRs) bind directly to the ECD of BRI1 to activate its ICD, thus conferring a BR-specific function[22,24–26]. After binding BRs, BRI1 interacts with BAK1 (BRI1 Associate Kinase 1) and SBI1 (Suppressor of *bri1 1*)[27–29]. This activates BRI1 to phosphorylate BSK1 (BR-Signaling Kinase 1) and BRI1-specific substrate (BKI1, BRI1-Kinase inhibitor 1)[30,31], which successively releases BKI1 and inactivates the negative regulator, BIN2 (Brassinosteroid Insensitive 2)[32,33]. This then activates BSU1 (bri1 Suppressor 1) and BES1/BZR1 (bri1 EMS-Suppressor 1)/(Brassinozole Resistant 1) transcription factors to regulate plant growth and development[34–36]. Finally, activated BES1/

BZR1 regulates the expression of numerous BR responsive genes[37,38].

*ems1* null mutants appear almost normal, but lack pollen, while *bri1* null mutants display extreme dwarfism with almost normal pollen[10,11,39], implying their non-overlapping biological functions[10,11,21,40]. In this work, we show that the BRI1 and EMS1 intracellular domains are functionally exchangeable. We find that expression of *BRI1* in the *EMS1* expression domain and co-expression of *EMS1* and *TPD1* in the *BRI1* expression domain can partially complement *ems1* and *bri1* mutants, respectively, suggesting that they can activate the same downstream components. We show that EMS1 and BRI1 originated in early land plants and flowering plants, respectively, and suggest a route for functional divergence of RLKs.

## Results

### The intracellular domains of EMS1 and BRI1 are interchangeable.
To identify potential downstream signaling molecules of EMS1 in the tapeta, we used a molecular complementation approach, considering sequence homology and evolutionary conservation to design domain swaps (Supplementary Fig. 1). The RLK family arose from a common ancestor and has since expanded by gene duplication and divergence[18,19]. Thus, RLKs might share similar downstream components even though they control very different biological functions. Therefore, the successful substitution of an ICD of one RLK with that of another RLK ICD could imply common targets. To test this notion, we fused the BRI1 ECD with the ICD of EMS1 and other RLKs to generate chimeric receptors that can perceive BRs to activate the intracellular function of each respective RLK (Fig. 1a). These engineered receptors were introduced into the plants under the control of *BRI1* promoter. We found that only the expression of $pro^{BRI1}::ECD^{BRI1}-ICD^{EMS1}$ (BRI1-EMS1), but not $pro^{BRI1}::ECD^{BRI1}-ICD^{PSKR1}$ (BRI1-PSKR1), nor any other $pro^{BRI1}::ECD^{BRI1}-ICD^{RLKs}$ (BRI1-RLKs), rescued *bri1* mutant phenotypes (Fig. 1a–c and Supplementary Figs. 1, 2a, b and 3a, b), consistent with the hypothesis that both EMS1 and BRI1 ICDs can activate common targets.

To test whether the BRI1-EMS1 chimera indeed activate BR-dependent targets, we exogenously applied brassinolide (BL) and brassinazole (BRZ), a BR biosynthesis inhibitor that blocks the production of BRs[41], to the transgenic seedlings. This showed that they had similar sensitivity to BR as that of wild type (WT) or transgenics expressing *BRI1* (Supplementary Fig. 3c–f). These results indicate that the chimeric BRI1-EMS1 is able to functionally substitute for BR receptors, probably acting through similar downstream molecules as BRI1. Accumulation of dephosphorylated BES1/BZR1 upon BR treatment is indicative of active BR signaling[35,36], and we detected higher accumulation of dephosphorylated BES1 in *BRI1-EMS1/bri1* transgenics compared to *bri1* mutants. To test whether the accumulation of dephosphorylated BES1 is due to activation of the chimeric BRI1-EMS1 receptor by BRs, we treated seedlings with BL and found further accumulation of dephosphorylated BES1 in the *BRI1-EMS1/bri1* transgenics but not in *bri1* mutants (Fig. 1d). Additionally, the expression of other BR regulated genes was also restored compared to WT and *bri1* mutants (Supplementary Fig. 3g–i). Altogether, our results indicate that chimeric BRI1-EMS1 can functionally substitute for BRI1 when expressed in the *BRI1* expression domain.

As EMS1-TPD1 downstream signaling events are almost unknown in the tapeta, we asked whether the BRI1 ICD could replace that of EMS1 in the tapeta. BRI1 and PSKR1 ICDs were fused with the EMS1 ECD and the chimeric receptors were expressed in *ems1* null mutant backgrounds under the control of

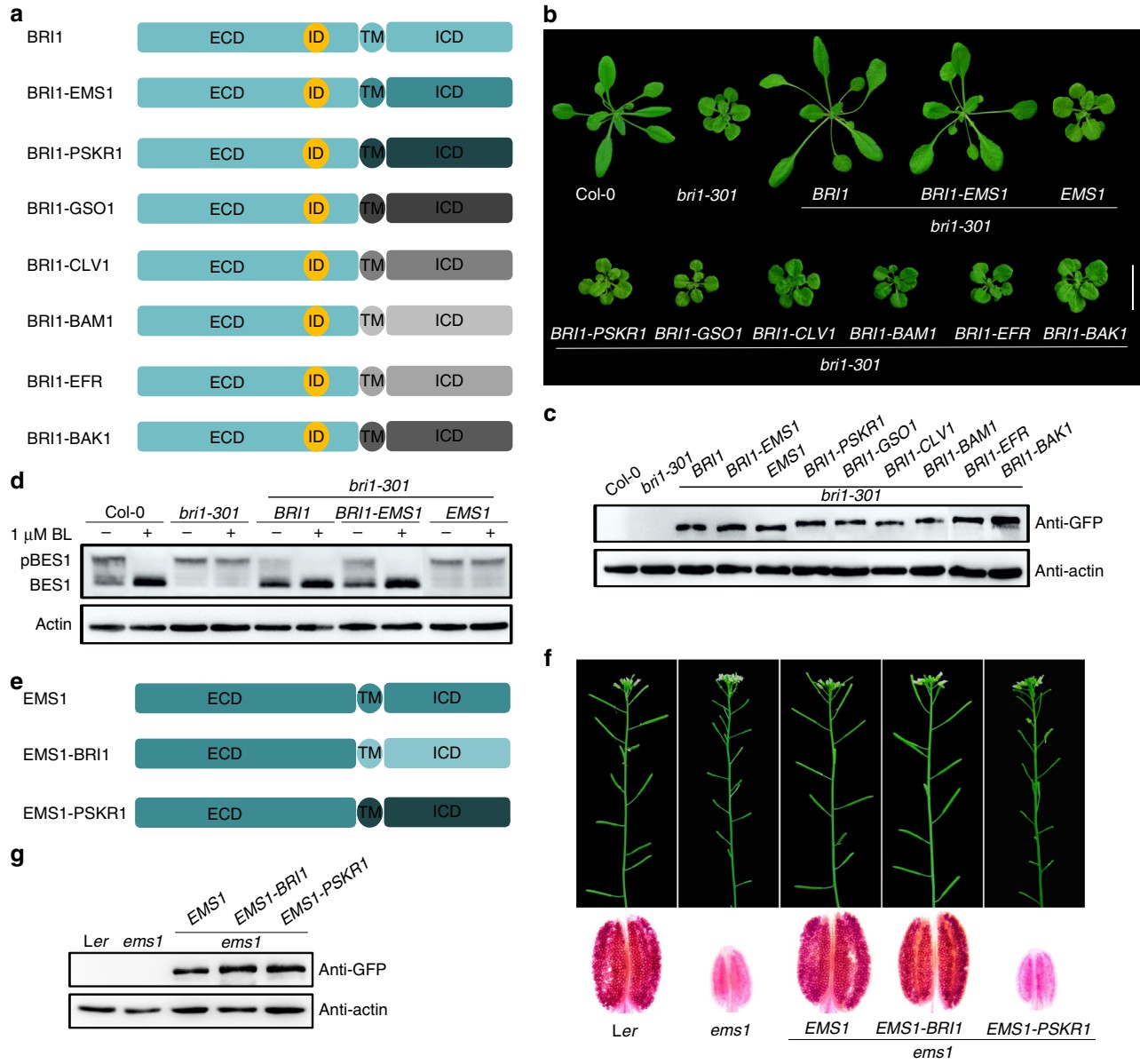

**Fig. 1** The intracellular domains of EMS1 and BRI1 are interchangeable. **a** Schematic diagram of the extracellular domain of BRI1 fused with the transmembrane domains and intracellular kinase domains of several LRR-RLKs that were labeled in different colors, respectively. **b** Phenotypes of 4-week-old transgenic lines expressing *BRI1*, *BRI1-EMS1*, *EMS1*, *BRI1-PSKR1*, *BRI1-GSO1*, *BRI1-CLV1*, *BRI1-BAM1*, *BRI1-EFR* and *BRI1-BAK1* under the *BRI1* promoter in *bri1-301* background. Scale bar, 2.0 cm. **c** Protein expression levels of the transgenes with GFP tag in the rosette leaves of the corresponding plants shown in **b** were detected with anti-GFP antibody. Actin served as the loading control. **d** Phosphorylated BES1 (pBES1) and dephosphorylated BES1 were detected with BES1 antibodies in the extracts of 10-day-old seedlings of the indicated genotypes. Where indicated, the plants were treated with 1 µM BL for 1 h before preparation of the extracts. Actin served as the loading control. **e** Schematic diagram of chimeric receptor kinases EMS1-BRI1, EMS1-PSKR1. The EMS1, BRI1 and PSKR1 protein structures were labeled in blue, light blue and dark blue, respectively. **f** Phenotypes of 6-week-old transgenic lines expressing *EMS1*, *EMS1-BRI1*, *EMS1-PSKR1* under the *EMS1* promoter in *ems1*. Primary inflorescences (top) and Alexander staining of pollen grains in mature anthers (bottom) showing the fertility phenotypes of the transgenic plants. **g** Protein expression levels of the transgenes with GFP tag in the inflorescences of the corresponding plants shown in **f** were detected with anti-GFP antibody. Actin served as the loading control

*EMS1* promoter. The expression of *pro^EMS1^::ECD^EMS1^-ICD^BRI1^* (EMS1-BRI1) but not *pro^EMS1^::ECD^EMS1^-ICD^PSKR1^* (EMS1-PSKR1) completely rescued the *ems1* mutant phenotypes (Fig. 1e–g and Supplementary Fig. 2c, d). This suggests that the signaling events that occur downstream of EMS1 may be activated by the BRI1 ICD in tapeta. Taken together, our results suggest that the ICDs of EMS1 and BRI1 are interchangeable.

**EMS1-TPD1 and BRI1-BRs receptor-ligand pairs can cross-complement.** In plants, EMS1-TPD1 and BRI1-BRs control completely different biological processes, yet the BRI1 and EMS1 ICDs are functionally interchangeable, implying that they might share common downstream signaling events. To test this hypothesis further, we individually expressed *EMS1* and *TPD1* under the control of *BRI1* promoter in Arabidopsis. Both failed to rescue *bri1* mutant phenotypes. However, co-expression of *pro^BRI1^::EMS1* and *pro^BRI1^::TPD1* (EMS1 & TPD1) together partially rescued the dwarf stature phenotype of the *bri1* mutant (Fig. 2a–c, e, f and Supplementary Figs. 4a–d and 5a). Yet, *pro^BRI1^::EMS1* and *pro^BRI1^::TPD1/bri1* co-expressing plants still had shorter

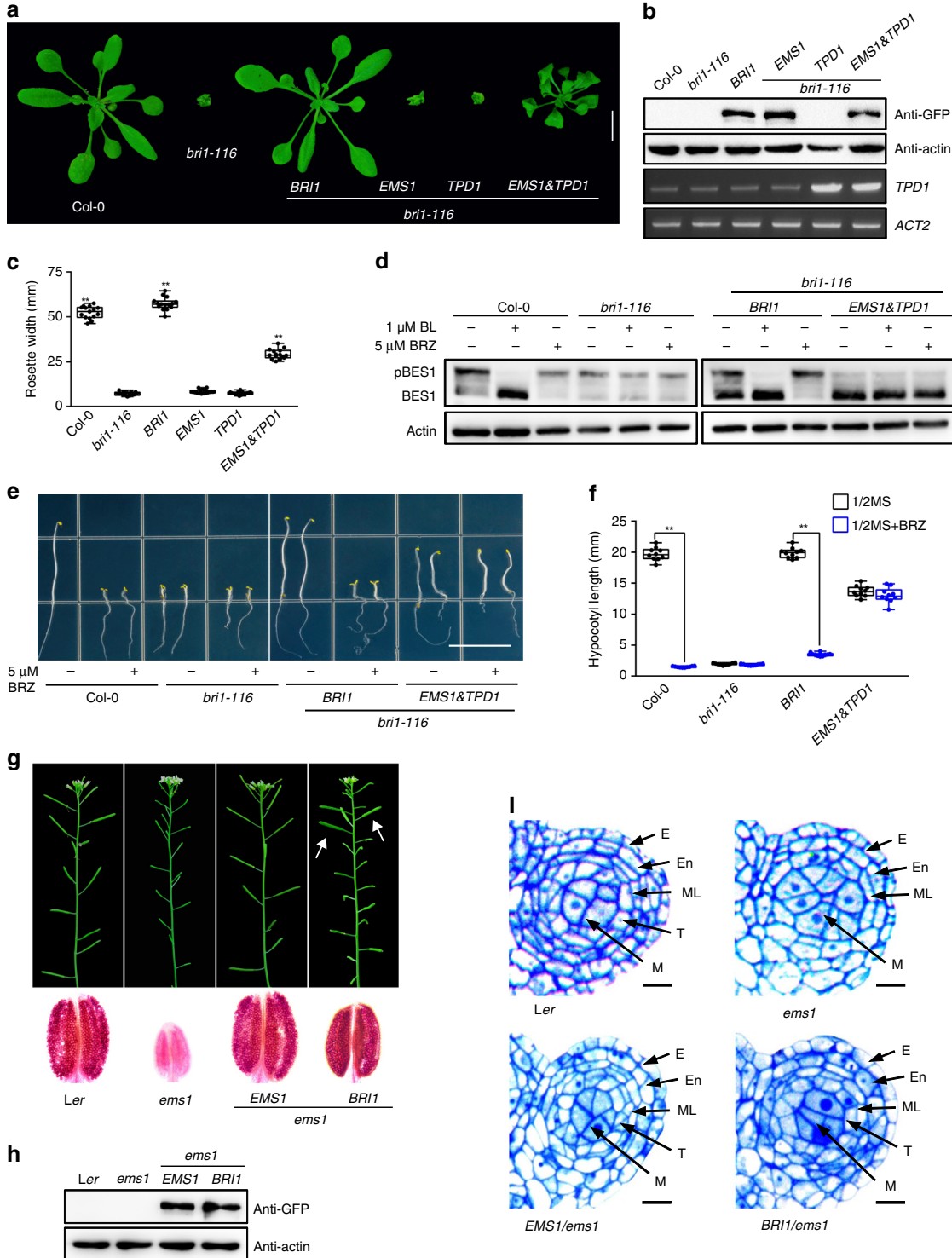

hypocotyls, smaller rosette diameters, curled and epinastic leaves, delayed senescence, reduced male fertility, and smaller siliques with fewer seeds compared to WT plants (Fig. 2a–f and Supplementary Figs. 4a–d and 5). Furthermore, $pro^{BRI1}::EMS1$ and $pro^{BRI1}::TPD1/bri1$ co-expressing plants were not sensitive to BRs, as was shown by treatment with BL and BRZ (Fig. 2d–f), confirming that the partial rescue was BR independent. To test if this functional recovery is correlated with activation of signaling components downstream of BRI1, we examined the accumulation and phosphorylation status of BES1 in the co-expression plants. Significantly, we detected accumulation of dephosphorylated

BES1 in the co-expression lines compared to $bri1$ mutants and this was independent of BRs (Fig. 2d and Supplementary Fig. 4a–d). We noticed a correlation between the accumulation of dephosphorylated BES1 and plant phenotypes, although it was non-linear (Fig. 2d and Supplementary Fig. 4a–d), possibly suggesting regulatory feedback. Nevertheless, our results indicate that BES1 can be activated independently of BRs by ectopic expression of EMS1-TPD1.

To further assess the similarity between ectopic EMS1-TPD1 signaling and native BR signaling, we used quantitative real-time PCR to analyse the expression of BR biosynthetic genes $CPD$ and

**Fig. 2** EMS1-TPD1 and BRI1-BRs receptor-ligand pairs can cross-complement. **a** The phenotypic recovery of *bri1-116* null allele by expressing *BRI1*, *TPD1*, *EMS1* independently and *EMS1* together with *TPD1* under the control of *BRI1* promoter. Scale bar, 1 cm. **b** Analyses of the expression levels of the transgenes in the rosette leaves of the corresponding plants shown in **a** Protein expression levels were detected with anti-GFP antibody. Actin served as the loading control. *TPD1* expression levels were detected by semi-quantitative RT-PCR. *ACT2* served as an internal control. **c** Quantification of the transgenic lines with the diameter of the rosette leaves in the whole plants grown for 4 weeks, *n* = 15 plants, *$P < 0.05$, **$P < 0.0001$ as one-way ANOVA with a Tukey's test. **d** Dephosphorylation of BES1 by both BRI1-BRs and EMS1-TPD1 signaling pathways. Expression of *BRI1*, or *EMS1* together with *TPD1* under the control of *BRI1* promoter rescued the dephosphorylation of BES1 in BR-dependent and BR-independent manner, respectively. BL brassinolide, BRZ brassinozole. Phosphorylated BES1 (pBES1) and dephosphorylated BES1 were detected with BES1 antibodies in the extracts of 10-day-old seedlings of the indicated genotypes. **e, f** Co-expression of *EMS1* & *TPD1* in *bri1-116* partially rescued the hypocotyl elongation of *bri1-116* mutant but did not restore the sensitivity to BRZ. 5-day-old dark-grown seedlings in 1/2 MS medium treated with or without 5 μM BRZ. Scale bar, 1.5 cm. *n* = 10 seedlings. **$P < 0.0001$ (two-way ANOVA with Sidak's test). **g** Anthers with or without pollen grains in the transgenic lines expressing *EMS1* or *BRI1* under the control of *EMS1* promoter in *ems1* background. Arrows, fertile siliques. **h** Protein expression levels of the transgenes with GFP tag were detected in the inflorescences of the corresponding plants shown in **g** with anti-GFP antibody. Actin served as the loading control. **i** Semi-thin sections of stage-5 anthers showing the normal anther cell differentiation in Ler, *proEMS1::EMS1* in *ems1* or *proEMS1::BRI1* in *ems1* as well as the abnormal anther cell differentiation in *ems1* (lack of tapetal cells). E epidermis, En endothecium, ML the middle layer, T tapetal cells, and M microsporocytes. Scale bars, 10 μm

*DWF4* or BR catabolic gene *BAS1* in the transgenic plants of *pro^BRI1^::EMS1* and *pro^BRI1^::TPD1/bri1* co-expression lines. *CPD* and *DWF4* were dramatically downregulated while *BAS1* was upregulated to resemble that of WT (Supplementary Fig. 5c–e). These data indicate that ectopic EMS1-TPD1 signaling can trigger a similar expression response to native BRI1-BR signaling. These results also suggest that BRI1-BR signaling is largely replaceable by EMS1-TPD1 when expressed in the *BRI1* expression domain and that similar downstream signaling events are activated.

If EMS1-TPD1 and BRI1-BRs do share key downstream events, then their replacement should be bidirectional. BRs are small hydrophobic chemicals, which may be difficult to restrict in between tapeta cell layers and microsporocytes in the manner of 20 kDa protein TPD1[12,42]. However, pollen is a rich source of endogenous BRs, meaning that it likely has high level of free BRs in the anthers[43]. We reasoned if BRs were enriched during pollen development, and if BRI1 was in the tapeta, it could access BRs to initiate BR signaling in tapeta, thus suppressing the *ems1* mutant phenotypes if the EMS1-TPD1 function can be replaced by BRI1-BR function. To test this hypothesis, we expressed the BRI1 receptor in tapeta by introducing the *BRI1* gene under the control of *EMS1* promoter into the *ems1* mutant plants. Indeed, the expression of *pro^EMS1^::BRI1* largely suppressed *ems1* mutant phenotypes (Fig. 2g–i and Supplementary Fig. 4e–f). Furthermore, we observed that the transgenic plants had viable pollen and normal seeds, although they had fewer and shorter siliques (Fig. 2g–i and Supplementary Fig. 4e, f). Significantly, *pro^EMS1^:: BRI1/ems1* plants had normal tapetal cells while *ems1* had none (Fig. 2i). Altogether, this suggests that EMS1-TPD1 is partially replaceable by BRI1-BRs in tapetum development.

**Dominant BR signaling mutants can modulate EMS1 signaling**. To further investigate the downstream components in EMS1-TPD1 signaling, we took advantage of two available dominant mutants *bzr1-1D* and *bes1-D* that suppress nearly all phenotypic defects of the *bri1* mutants. *bzr1-1D* and *bes1-D* are point mutation mutants, where both active (dephosphorylated) and inactive (phosphorylated) BES1/BZR1 accumulate, causing BR signaling to be amplified with or without BRs[35,36]. We reasoned that if EMS1-TPD1 functions through BES/BZR1, then the tapeta may develop without EMS1-TPD1 in these mutants. Thus, we crossed *bzr1-1D* and *bes1-D* mutants with *ems1* and *tpd1* single mutants and *serk1serk2* double mutants. We found that all double or triple mutants, *ems1bzr1-1D*, *ems1bes1-D*, *tpd1bzr1-1D*, *tpd1bes1-D*, *serk1serk2bzr1-1D* or *serk1serk2bes1-D*, exhibited normal tapeta, pollen and seeds (Fig. 3a, b and Supplementary Fig. 6), consistent with our hypothesis. However, the native *BZR1* and *BES1* are ubiquitously

expressed in plants and there is a possibility that these dominant effects are indirect. We thus expressed the *bzr1-1D* and *bes1-D* point mutant variants under the *EMS1* promoter in the *ems1* mutants, and found that they exhibited similar phenotypes as the dominant genetic mutants (Supplementary Fig. 6g, h), confirming that BZR1 and BES1 can indeed promote tapetum development when expressed in the *EMS1* expression domain.

BIN2 is a repressor of BRI1-BR signaling, and *bin2-1D* mutant phenotypes resemble those of *bri1* mutants[32]. To determine if BIN2 can also suppress the EMS1-TPD1 signaling pathway, we co-expressed *TPD1* and *EMS1* under the *BRI1* promoter in the *bin2-1D* mutant background. We found that the *EMS1* with *TPD1* expression could not overcome the inhibitory effects of the *bin2-1D* mutants (Fig. 3c, d), suggesting that BIN2 can also act downstream of EMS1-TPD1.

**EMS1 and BRI1 signaling complementarily control the *Arabidopsis* life cycle**. To further understand the role of EMS1-TPD1 and BRI1-BRs in the control of plant development, *bri1-116* null mutants were crossed to *ems1* mutants to generate *bri1-116ems1* double mutants. We found that the double mutants had an additive effect of both single mutant phenotypes, resulting in dwarf plants lacking tapeta and mature pollen (Supplementary Fig. 7a). This suggests that EMS1-TPD1 and BRI1-BRs independently control separate developmental processes in Arabidopsis. To further test their independence, we expressed *pro^BRI1^:: EMS1* in *ems1* mutants and found that it did not suppress *ems1* mutant phenotypes (Supplementary Fig. 7b, c). Consistently, we observed stronger expression of GFP in the tapetal cell layer of transgenic plants expressing *pro^EMS1^::EMS1::GFP* than in transgenic plants expressing *pro^BRI1^::EMS1::GFP* (Supplementary Fig. 7d), suggesting little or no BRI1 accumulation in the tapeta.

Interestingly, null *bes1/bzr1* family mutants are dwarf and lack pollen resembling *ems1bri1* double mutants[44]. This is consistent with a role for the native BES1/BZR1 family controlling tapetal cell fate downstream of EMS1-TPD1.

**EMS1-TPD1 are conserved across land plants**. To investigate the conservation of EMS1-TPD1 signaling in terrestrial plants, we studied *Physcomitrella patens*, one of the earliest land plant species lacking true tapeta[45]. We found that EMS1 and TPD1 are present in completely sequenced genomes of all land plants but not in alga (some examples were shown in Supplementary Fig. 8)[45–51]. *P. patens* (*Pp*) had EMS1 (PpEMS1) and TPD1 (PpTPD1) but not BRI1 or BRI1-likes (BRLs)[21,26,45,51–54]. There were six PpEMS1 and one PpTPD1 in *P. patens* (Supplementary Fig. 8). To determine whether they have functional ICDs, we

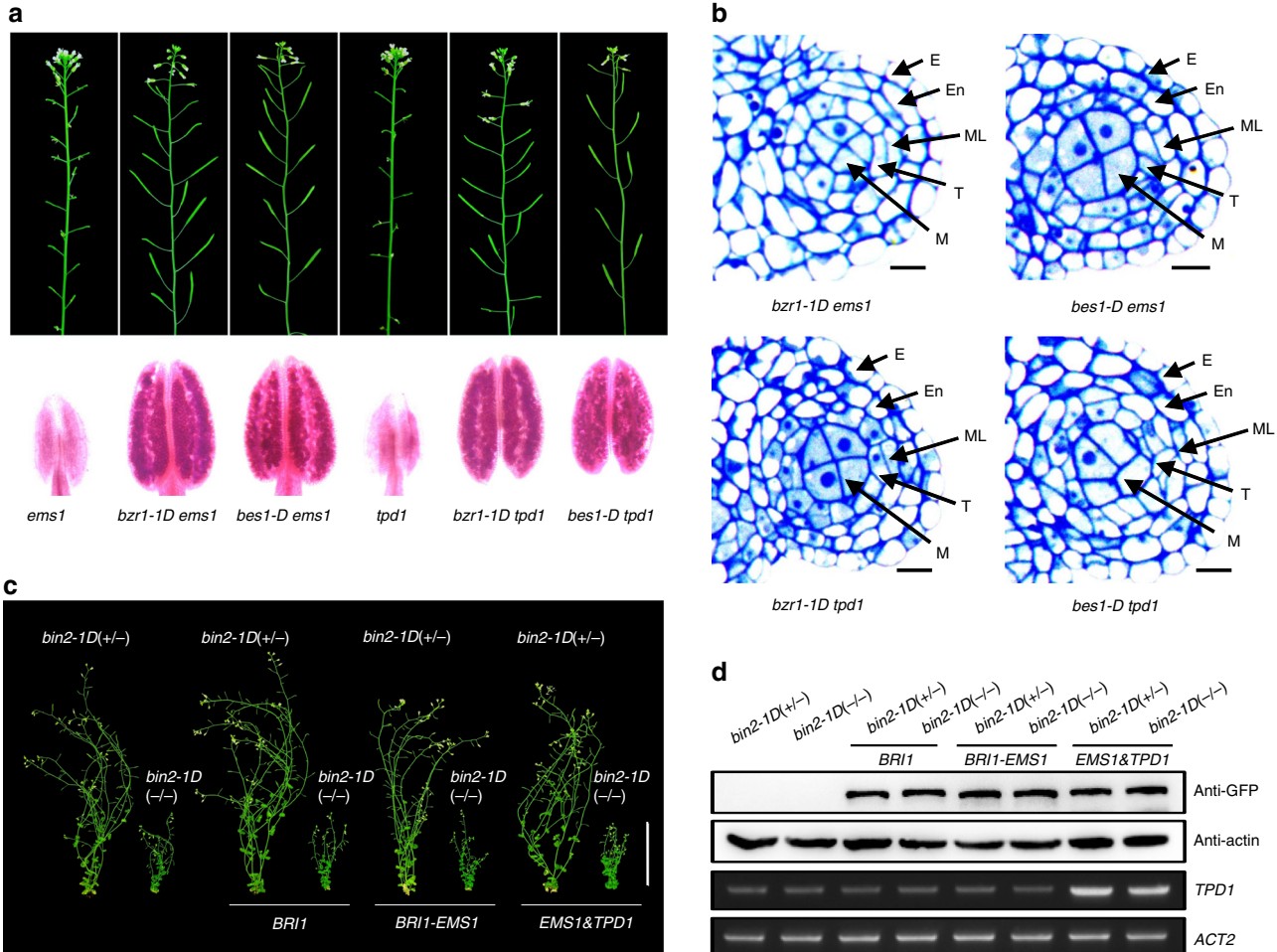

**Fig. 3** Activated transcription factors of BRI1-BRs signaling can rescue *ems1* and *tpd1* mutants. **a** *bes1-D* and *bzr1-1D* rescued the male sterile phenotypes of *ems1* and *tpd1* mutants. Primary inflorescences (top) and Alexander staining of pollen grains in mature anthers (bottom) showing the fertility phenotypes of *bzr1-1Dems1*, *bes1-Dems1*, *bzr1-1Dtpd1* and *bes1-Dtpd1* double mutants. **b** Semi-thin sections of stage-5 anther lobes showing four somatic cell layers, including epidermis (E), endothecium (En), the middle layer (ML) and tapetum (T), as well as reproductive microsporocytes (M) in the center of *bzr1-1Dems1*, *bes1-Dems1*, *bzr1-1Dtpd1* and *bes1-Dtpd1* double mutants, respectively. Scale bars, 10 μm. **c** BIN2 inhibited EMS1-TPD1 signaling. Phenotypes of 7-week-old transgenic plants expressing *BRI1*, *BRI1-EMS1*, *EMS1* and *TPD1* under the control of *BRI1* promoter in *bin2-1D* (+/−) and *bin2-1D* (−/−) plants. Scale bar, 5 cm. **d** Analyses of the expression levels of the transgenes in the rosette leaves of the corresponding plants shown in **c**. Proteins with GFP tag were detected with anti-GFP antibody. Actin served as a loading control. *TPD1* expression was detected by semi-quantitative RT-PCR. *ACT2* served as an internal control

fused the BRI1 ECD with the six PpEMS1 ICDs. All of these chimeric constructs suppressed the phenotypes of *bri1* mutants when expressed under the *BRI1* promoter (Supplementary Fig. 9). Furthermore, the expression of *PpEMS1-1* and *PpEMS1-2* under the control of the *EMS1* promoter completely rescued *ems1* phenotypes. Meanwhile, *PpTPD1* expression under the *TPD1* prompter in *tpd1* mutant completely recued *tpd1* phenotypes (Fig. 4a, b). Importantly, co-expression of either *PpEMS1-1* or *PpEMS1-2* with *PpTPD1* under the *BRI1* promoter in *bri1* mutants partially suppressed the *bri1* dwarf phenotype and caused accumulation of dephosphorylated BES1 (Fig. 4c–e). As expected, these transgenic seedlings were not sensitive to BRZ treatment, showing no retardation in hypocotyl growth (Fig. 4f–g).

Altogether, these findings suggest that EMS1 and TPD1 have been able to form a functional pair for at least 400 million years[46,55]. Since BRI1 and BRLs were not found in *P. patens* or other early land plants, such as liverworts[46], this implies that BR signaling emerged after EMS1-TPD1 signaling. Phylogenetic analysis revealed that BR receptors shared common ancestors with EMS1 (ref. [56]).

Significantly, BRI1 was not found in any non-flowering plants with completely sequenced genomes, including liverworts, mosses, lycophytes, ferns and gymnosperms[45–50,56,57], implying that BRI1 was rapidly neofunctionalized in angiosperms, and likely attributing to the adaptive advantage of flowering plants. Altogether, we propose that expansion of BRI1/EMS1 signaling may have accompanied land plant evolution. EMS1 may be important for all land plants whereas BRI1 appears only essential in flowering plants (Fig. 4h).

## Discussion

Cell-to-cell communication is essential for cell differentiation and development, which is largely controlled by cell surface receptors that perceive and transmit various signals from the outside into the inside of the cell to control a wide range of physiological and developmental processes[4]. RLKs are the largest family of cell surface receptors in land plants. In Arabidopsis alone, there are more than 600 RLKs, accounting for 2.5% of all coding proteins, 60% of all kinases and nearly all

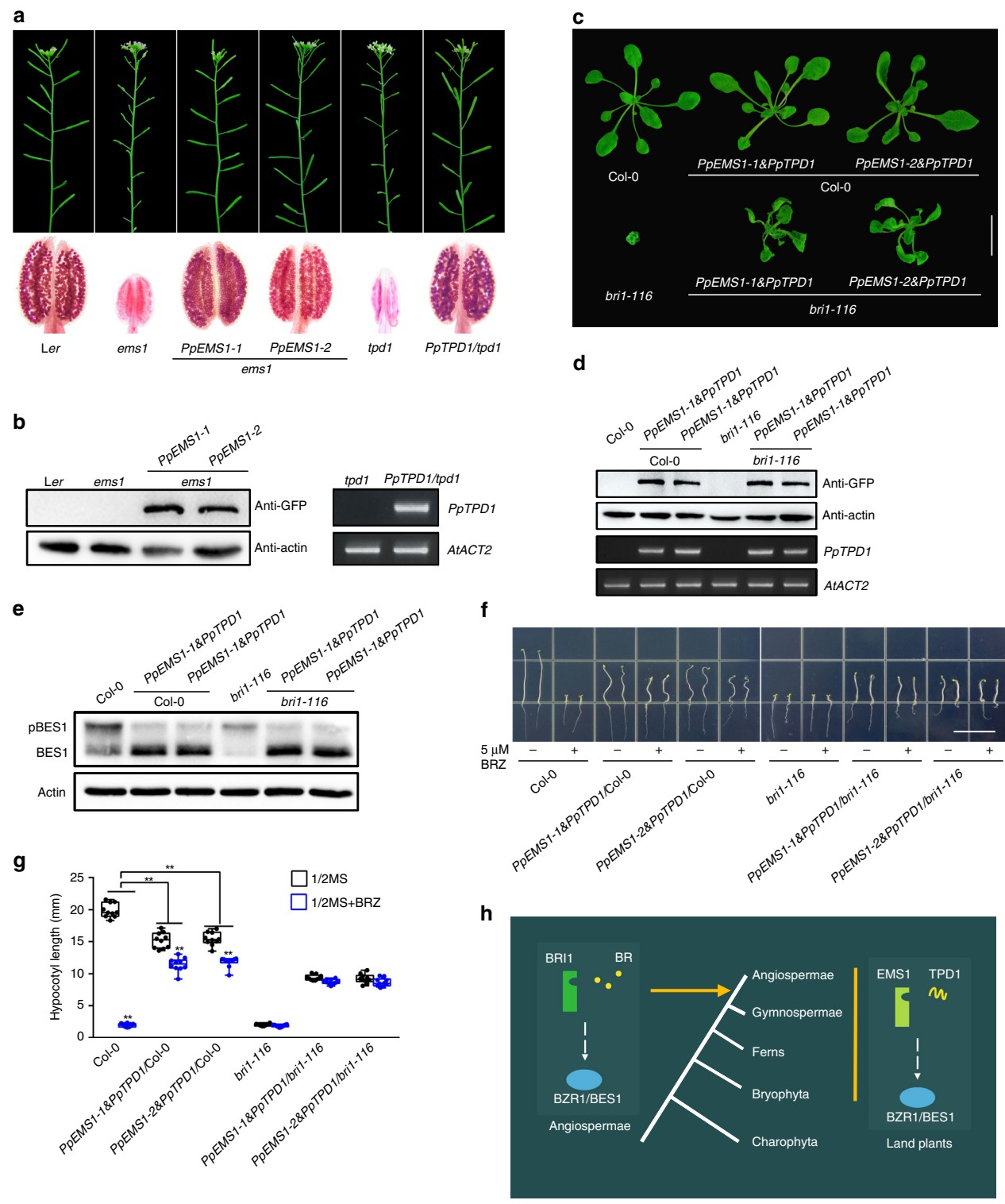

transmembrane kinases[18,19,58,59]. Likely due to redundancy or lethality, most of them have unknown functions; however, some of them have been studied, but only a few of them are well characterized. Currently, forward genetics is still the most effective approach to study their signaling pathways. Benefiting from numerous viable genetic mutants, BRI1 has one of the best known signaling pathways while EMS1 has one of the least, probably because it is essential (genetic mutants are lethal)[10,11,26,60]. There is a need for new strategies to address this problem.

In the RLKs, the ECDs evolve faster than ICDs, so it is possible that ECDs have acquired the ability to perceive distinct signals, while the ICDs still target similar downstream molecules to control apparently diverse biological processes[18,59]. This allows us to swap the ECDs and ICDs to molecularly complement mutants between the RLKs with and without known signaling components. When complementation is phenotypically substantial, partly or completely in both directions, the tested RLKs are likely able to activate a common set of downstream molecular components when present in the reciprocal expression domains. In

**Fig. 4** PpEMS1 and PpTPD1 (from moss *Physcomitrella patens*) function as Arabidopsis EMS1 and TPD1 in Arabidopsis, respectively. **a** *PpEMS1* and *PpTPD1* completely restored the phenotypes of *ems1* and *tpd1*, respectively. **b** Analyses of the expression levels of the transgenes in the inflorescences of the corresponding plants shown in **a**. Proteins with GFP tag were detected with anti-GFP antibody. Actin served as the loading control. *PpTPD1* expression levels were detected by semi-quantitative RT-PCR, *ACT2* served as an internal control. **c** Co-expressing *PpEMS1* and *PpTPD1* under the *BRI1* promoter partially rescued *bri1* phenotypes. Phenotypes of 4-week-old Col-0, *PpEMS1-1 & PpTPD1* and *PpEMS1-2 & PpTPD1* in Col-0 and *bri1-116* mutants were shown, respectively. Scale bar, 2 cm. **d** Analyses of the expression levels of the transgenes in the inflorescences of the corresponding plants shown in **c**. Proteins with GFP tag were detected with anti-GFP antibody. Actin served as the loading control. *PpTPD1* expression levels were detected by semi-quantitative RT-PCR. *ACT2* served as an internal control. **e** Co-expressing *PpEMS1* and *PpTPD1* under the *BRI1* promoter in Col-0 and *bri1-116* induced BES1 dephosphorylation. Phosphorylated BES1 (pBES1) and dephosphorylated BES1 were detected with BES1 antibodies in the extracts of 10-day-old seedlings of the indicated genotypes. Actin served as the loading control. **f, g** 5-day-old dark-grown seedlings in 1/2 MS medium with or without 5 μM BRZ. Scale bar, 1.5 cm. $n = 10$ seedlings. **$P < 0.0001$ (two-way ANOVA with Sidak's test). Co-expressing *PpEMS1-1 & PpTPD1* and *PpEMS1-2 & PpTPD1* in Col-0 or *bri1-116* showed less sensitivity or insensitivity to BRZ, respectively. **h** A proposed model illustrating that EMS1 and BRI1 have evolved distinct extracellular domains to control different biological processes but can act via a common intracellular signaling pathway

---

our case, we made chimeric EMS1 and BRI1 receptors using their respective ECDs and ICDs, but theoretically this approach is not limited to the RLKs and could be applied to other diverged protein pairs.

It should be noted that our approach relies upon the ectopic expression of EMS or BRI1 in reciprocal expression domains. It is therefore conceivable that phenotypic rescue and correlated molecular events may not reflect the native function of the protein. For example, it is possible that EMS1-TPD1 mediated activation of BES1 occurs due to ectopic expression of EMS1 in the BRI1 expression domain. Likewise, dominant mutants of BES1/ BZR1 expressed in the tapetum might conceivably lead to effects that are not typical of native EMS1 signaling. Nevertheless, given that the complementation is reciprocal and that higher order BES1/BZR1 loss-of-function mutants demonstrate fertility defects[44], our analyses suggest that EMS1 shares common signaling components with BR signaling. In this cascade, the cell surface receptor EMS1 binds with small protein ligand TPD1 to activate downstream components[7,12]. We propose that this activates the BZR1 and BES1 transcription factors to regulate the expression of target genes to determinate the tapetum cell fate. Several lines of evidence support this model: (i) the functional domains of EMS1 and BRI1 can be exchangeable; (ii) EMS1-TPD1 and BRI1-BR signaling can trigger similar molecular response when expressed in each other's expression domains; and (iii) dominant mutants *bzr1-1D* and *bes1-D* suppress both *bri1* and *ems1* mutants. BRI1-BR signaling regulates almost all aspects of plant growth and development[21,35,37,39], while EMS1-TPD1 signaling only controls tapetum determination[6,7,10,11], therefore it may be surprising that such different biological processes could be controlled by similar molecular components. However, it may be the case that it is more economical for plants to use limited molecules to regulate a variety of plant development and cope with complex and variable internal and external environments.

A significant question is how did EMS1 and BRI1 evolve? One possibility is that EMS1 and BRI1 were duplicated and diverged from a common ancestor[61–63]. The ancestral duplicates have similar but tapetum-non-specific expression patterns and functions, which generates a potential adaptive conflict (optimizing one function compromising the other) requiring resolution[64]. In this case, a steroid binding site is thought to have gradually emerged in ancestral BR receptors[56]. When the ligand switches from a protein TPD1 of 20 kDa to a chemical BR of only about 0.5 kDa[12,42], it could have allowed rapid evolution in the ECDs of ancestral BRI1, allowing BR activity to coevolve with BR-binding affinity in the ancestral BRI1. In complex tissues and organs, small ligands such as BRs may be easier to transport. Conversely, a large protein ligand such as TPD1 could be more easily confined to the space between the tapetal cell layer and the pollen mother cell than a small ligand that may diffuse to the neighboring cells

resulting in functional imprecision, thus favoring the large ligand for communications between two cell layers[12]. Therefore, we propose that EMS1 perceives TPD1 in tapeta while BRI1 binds BRs elsewhere to control whole plant development in a complementary manner acting via a common downstream signaling pathway[6,10,11,21]. As a result, both EMS1 and BRI1 became essential in angiosperms[7,11,21,40,65]. Future study of the functional divergence of EMS1 and BRI1 could boost the understanding of the function and evolution of other RLKs.

Our study on EMS1 and BRI1 suggests that RLKs can perceive distinct ligands but trigger a common signaling pathway to control diverse biological and physiological processes. This suggests ligand-driven coevolution of RLKs in their ECDs could have resulted in diverse ECDs but conserved ICDs in RLKs.

## Methods

**Plant materials and growth conditions**. The *Arabidopsis thaliana* Landsberg *erecta* (L*er*) and Columbia (Col-0) ecotypes were used as wild type (WT) control in this study. *ems1* and *tpd1* mutants are in the L*er* background. *bri1-301*, *bri1-116*, *det2*, *bin2-1D*, *serk1* (SALK_044330), *serk2* (SALK_058020) and *bzr1-1D* alleles are in Col-0 background. *bri1-116* is in *Enkheim-2 (En-2)*. Seeds were germinated on either ½ Murashige & Skoog (MS) medium, then transferred to the soil or directly planted in the soil. Plants were grown under long daylight conditions (16-h light/8-h dark cycles). Seedlings for measurement of BR-dependent responses were germinated on ½ MS medium with or without BL (Solarbio) or BRZ in the dark or in the light. Pictures of the plates were then taken for measurement of root length and hypocotyl length using ImageJ software.

**Generation of constructs and transgenic plants**. For genetic analyses, the *BRI1*, *EMS1* and *TPD1* promoters were individually introduced into the *pCHF3* (kanamycin selection) and *pCAMBIA1300* (hygromycin B selection) plasmids with or without GFP to construct backbones listed in Supplementary Table 2. Primers for generating these backbones were included in Supplementary Table 1. The cDNA sequences of *BRI1*, *EMS1*, *TPD1*, *PpTPD1* and the genomic sequences of *PpEMS1s* were inserted into the vectors' backbones to complement the mutants as indicated in the text or figures. Overlapping PCR was used to replace the ECD (extracellular domain) or ICD (intracellular domain) to generate $ECD^{BRI1}:ICD^{RLKs}$, $ECD^{EMS1}:ICD^{RLKs}$ and $ECD^{BRI1}:ICD^{PpEMS1s}$ chimeric genes. The genomic sequences of *bes1-D* and *bzr1-1D* were cloned to activate BRI1 or EMS1 signaling pathway. They were then expressed in the Col-0, *bri1-116*, *bri1-301*, *ems1*, *det2*, or *bin2-1D* as indicated in the figures or text. All above constructs were transferred into plants via *Agrobacterium* (GV3101)-mediated transformation using the method described elsewhere[66]. The transformants were then screened on ½ MS with 50 μg/ml kanamycin or 40 μg/ml hygromycin B. Double transgenic plants were produced by crossing, followed by a screen on ½ MS with 50 μg/ml kanamycin and 40 μg/ml hygromycin B. Primers and constructs used in this study were given in Supplementary Tables 1 and 2, respectively. Molecular genotyping was performed using the primers listed in Supplementary Table 1. At least 20 independent transgenic lines were obtained from each construct.

**Protein extraction and immunoblot analysis**. Total protein extracts were extracted from plant tissues with 2× SDS buffer (100 mM Tris, pH 6.8, 4% [w/v] SDS, 20% [v/v] glycerol, 0.2% [w/v] bromophenol blue, 2% [v/v] β-Mercaptoethanol), separated on SDS-PAGE gel, and then transferred to a Nitrocellulose membrane (Millipore). Anti-GFP antibodies (1:1000 dilution, Transgen, HT801) were used to detect GFP fusion proteins and BES1 antibodies (1:3000 dilution, kindly provided by J. Li, Lanzhou University, China) were used to detect the phosphorylation status of BES1. The equal loading was judged by Actin (1:1000 dilution, Abbkine A01050).

**Pollen staining and anther semi-thin sections**. To observe viable pollen grains, anthers prior to dehiscence were fixed in Carnoy's fluid (alcohol: chloroform: acetic acid = 6:3:1) for 2 h. Then, the anthers were dissected and stained with Alexander's solution at 50 °C for 48 h[67]. Anthers were photographed under a compound microscope. For the anther structure study[67], dissected floral buds and inflorescences were fixed in 4% (v/v) glutaraldehyde in 0.1 M PBS buffer (pH 6.8) and 0.02% (v/v) Triton X-100 overnight at 4 °C. Samples were washed 4 times for 15 min each in 0.1 M PBS (pH 6.8) and then fixed in 1% $OsO_4$ for 4 h. They were then dehydrated in a graded ethyl alcohol (20% increments) and embedded in Spurr's resin. Semi-thin (0.5 μm) sections were made by using a Leica EM UC7 Ultra-microtome (Leica Microsystems) and were stained with 0.25% of Toluidine Blue O. The images were photographed using an optronics digital camera. Tapetal cells and microsporocytes were determined under the microscope in the central sections of the anthers.

**Microscopy analysis**. Images of pollen staining and semi-thin sections were photographed under a Leica microscope equipped with a digital camera[12]. For confocal microscopy analysis, samples were observed under a Leica TCS SP5 laser scanning confocal microscope. They were then mounted in water and observed with a ×20 lens or a ×40/1.1 water immersion objective lens. A 488-nm laser was used to excite GFP and chlorophyll. The emission was captured using PMTs set at 505–530 nm and 644–719 nm for GFP and chlorophyll, respectively.

**qRT-PCR analysis**. Total RNA was isolated from the leaves of 4-week-old plants or 6-week-old inflorescences using a HiPure Plant RNA Mini Kit (Magen, R4151-02) according to the protocol provided by the manufacturer. First-strand cDNA was synthesized from 1 μg of total RNA using M-MLV First Strand cDNA Synthesis Kit (Omega, TQ2501-02). The qRT-PCR was performed using ChamQ$^{TM}$ SYBR qPCR Master Mix (Vazyme, Q311-00) to detect the transcript levels of genes. *AtACTIN2 (AtACT2)* was used as an internal control for qRT-PCR. The primers used are listed in Supplementary Table 1.

**Statistical analysis**. Statistical analysis was performed using one-way analysis of variance (ANOVA), two-way ANOVA and Tukey's test or Sidak's test, as implemented in GraphPad Prism 6.0 (GraphPad Software, http://www.graphpad.com).

**Sequence data**. Sequence data for this article can be found in the TAIR (https://www.arabidopsis.org/), Phytozome 12 database (https://phytozome.jgi.doe.gov/pz/portal.html#), congenie.org (http://congenie.org/) and FernBase (https://www.fernbase.org/) under the following accession numbers (all genes are from *Arabidopsis thaliana* except otherwise indicated):

BRI1, AT4G39400; EMS1, AT5G07280; TPD1, AT4G24972; BRL1, AT1G55610; BRL3, AT3G13380; BRL2, AT1G14000; PSKR1, AT2G02220; GSO1, AT4G20140; CLV1, AT1G75820; EFR, AT5G20480; BAM1, AT5G65700; BAK1, AT4G33430; CTR1, AT5G03730; BIN2, AT4G18710; BES1, AT1G19350; BZR1, AT1G75080; ACT2, AT3G18780; CPD, AT5G05690; DWF4, AT3G50660; BAS1, AT2G26710; SERK1, AT1G71830; SERK2, AT1G34210; OsMSP1, LOC_Os01g68870.1; AmEMS1, evm_27.model.AmTr_v1.0_scaffold00009.24; PaEMS1, MA_1913g0010; AfEMS1, Azfi_s0017.g014644; ScEMS1, Sacu_v1.1_s0054.g014268; SmEMS1, Sm99902; PpEMS1-1, Pp3c1_41620V3.1; PpEMS1-2, Pp3c14_16840V3.1; PpEMS1-3, Pp3c22_12040V3.1; PpEMS1-4, Pp3c19_18410V3.2; PpEMS1-5, Pp3c17_21540V3.3; PpEMS1-6, Pp3c1_16110V3.4; MpEMS1, Mapoly0011s0213.1; OsTPL1A, LOC_Os12g28750.1; AmTPD1, evm_27.model.AmTr_v1.0_scaffold00047.41; PaTPD1, MA_10427288g0010; AfTPD1, Azfi_s0003.g008001; ScTPD1, Sacu_v1.1_s0032.g010785; SmTPD1, Sm113463; PpTPD1, Pp3c22_22420V3.1; MpTPD1, Mapoly0020s0056.1. AT: *Arabidopsis thaliana*; Os: *Oryza sativa*; Am, *Amborella trichopoda*; Pa, *Picea abies*; Af, *Azolla filiculoides*; Sc, *Salvinia cucullata*; Pp: *Physcomitrella patens*; Sm: *Selaginella moellendorffii*; Mp: *Marchantia polymorpha*.

**Reporting summary**. Further information on research design is available in the Nature Research Reporting Summary linked to this article.

## Data availability

The authors declare that all data supporting the findings of this study are available within the manuscript and its supplementary files, or are available from the corresponding authors upon request. The source data underlying Figs. 1–4 and Supplementary Figs. 21–9 are provided as a Source Data file.

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

## Acknowledgements

We thank H. Ma for *ems1*, J. Li (U. Michigan) for *bin2-1D*, Z.Y. Wang for *bzr1-1D*, Y. Yin for *bes1-D* and J. Li (Lanzhou University) for *serk1serk2* (*serk1/2*) and BES1 antibodies. Financial support to G.W. from Shaanxi Normal University, Chinese Ministry of Education (313034) and (20130202110007); Fundamental Research Funds for Central Universities to G.W. (GK201101005), B.Z. (2017CB2004) and Q.B. (2017TS032); Grants from the Chinese National Foundation of Science to G.W. (31270324 and 31741014), and postdoctoral fellowships from Chinese National Foundation of Science to H.R. (2012M521740) and G.S.L. (2012M521739).

## Author contributions

G.W. and X.S. conceived and guided the study. B.Z., Q.B., X.S., and G.W. designed the experiments. Q.B., B.Z., L.W., H.L., and Y.L. performed the experiments. B.Z., Q.B., L.W., H.L., Y.L., G.L., W.X., H.R., X.S., and G.W. analyzed the data and interpreted the results. G.W. wrote the manuscript, and the other authors read and edited the manuscript.

## Additional information

**Competing interests:** The authors declare no competing interests.

