## [Peer Review File · Nature Communications]

Reviewers' comments:

Reviewer #1 (Remarks to the Author):

Zheng et al. present results from the analysis of the EMS1 and BRI1 receptor-like protein kinases using chimeric proteins in mutant backgrounds, and other experiments. These results are suggestive of structural similarity between BRI1 and EMS1, but do not provide sufficient evidence about the molecular mechanisms of the EMS1 protein.

The authors used "functional conservation" to refer to structural similarity of the ICDs of these proteins, as discussed in detail below. The results presented are not sufficient to provide new insights into the function of the EMS1-TPD1 signaling module.

It is possible that the BRI1-EMS1 chimera functions in the transgenic plants in a way similar to the BRI1 native protein; this is supported by the partial increase of the non-phosphorylated BES1 protein (in the leaves?). The genetic results also support the idea that the EMS1-BRI1 chimera can function like EMS1 protein in transgenic plants, but the downstream factors for this pathway is unknown, because no experiments were performed to detect changes in downstream proteins in the anther.

Therefore, the results are too preliminary to make firm conclusions regarding the mechanisms of EMS1.

The following specific comments must also be addressed before the manuscript can be published.

1. For the transgenic plants shown in Figure 1, information on the number of transgenic lines generated for each construct should be provided; in addition, the expression levels of the transgene, and the corresponding phenotypes for at least two independent transgenic lines should be presented, to avoid the possibility that some of the phenotypes and other properties of the transgenic plants might be caused by unknown mutations generated during the transformation process.
2. In the description of the phenotypes of the transgenic lines (from perhaps a single line here), some differences between BRI1 and BRI1-EMS have been overlooked. For example, the hypocotyl lengths of the BRI1-EMS seedlings in the absence of BRZ are less than those of the transgenic BRI1 seedlings. The description should be accurate and truly reflect the results.
3. In addition, in Figure 2, the hypocotyl lengths of seedlings expressing both EMS1 and TPD1 are intermediate between Col-0/BRI1 transgenic and the *bri1* mutant, indicating that the EMS1/TPD1 transgenes can only partially rescue the mutant phenotypes. Again the description should be revised.
4. Also, in Figure 2, although BRI1 was able to promote the formation of four layers of somatic cells, the EMS1 transgenic anther shows more cell layers in comparison. Furthermore, the fertility of the BRI1 plants is less than that of the EMS1 plants. Therefore, the ability of the EMS1 and BRI1 genes to rescue the *ems1* mutant is somewhat different. The description should be consistent with this fact.
5. In Figure 3, the tissues used for western experiments should be explicitly stated. Were leaves? Were they flowers? The phosphorylation state of BES1 protein in the plants with the EMS1 and TPD1 transgenes is intermediate between those of the BRI1 transgenic plants with or without BL added. These results suggest that although EMS1 ICD has a similar function to that of BRI1, but it is weaker.
6. Also, in Figure 3, the transcriptome analyses clearly showed that the EMS1-TPD1 transgenic plants have distinct sets of both up-regulated and down-regulated genes. Thus these genes do not act like the BRI1 gene. This should be clear.
7. The phylogenetic tree in Figure 4f is inaccurate, because lycophytes are not monophyletic (do not form a clade or branch): ferns are sister to seed plants and lycopods form a separate branch (Lycophytes form two branches). The situation for bryophytes is less clear. Some studies support monophyly of bryophytes, but others do not (with two or three branches). Also, there are several groups (branches) within charophytes, and some are more closely related to land plants than

others. Therefore, charophytes are also not monophyletic.

8. The phylogenetic analyses shown here in Supplemental Figures 1 and 6 should be described with more details, including the parameters for sequence retrieval, for alignment, and bootstrap values should be obtained and shown on the trees.

9. The writing of the manuscript contains a number of misconceptions and misstatements about interpretation of results and evolutionary conservation.

For example (there are more, sometimes similar situations):

9a. The use of transgene (including chimeric genes) in a mutant background can generate situations that suggest one gene might be able to act to substitute another gene. For example, the ICD of EMS1 could substitute that of BRI1 in transgenic plants. However, these results do not mean that the two genes being tested have evolutionarily conserved functions. The words "evolutionarily conserved functions" refer to their functions in the wild-type plants of the native species, not in transgenic plants generated in the lab (even in a heterologous plant).

In fact, these two genes, EMS1 and BRI1, clearly have distinct native functions as indicated by their dramatically different mutant phenotypes. So, they should not be described as having "conserved functions".

9b. In addition, rescue (partial rescue) of a mutant phenotype does not necessarily mean that the mutated gene and the rescuing gene are evolutionarily conserved. In fact, sometimes, a downstream gene can rescue a mutant in the upstream gene, when the two genes encode proteins of dissimilar sequences. For example, here BRI1 and BRI-EMS1 fusion were shown to partially rescue a *det2* mutant, but BRI1 and EMS1 clearly do not have conserved functions with respect to DET2.

The correct interpretation is that the EMS1 ICD has sufficient sequence and (probably) structural similarity to that of BRI1 to be able to partially substitute for the BRI1 when the EMS1 ICD is fused to the ECD of BRI1. These and other results in the manuscript do not indicate evolutionarily conserved functions, only structural similarity. The structural similarity would allow the chimeric protein to interact with some of the downstream factors of BRI1, but this is not sufficient to indicate that EMS1 normally interacts with such downstream factors.

The writing related to such experiments need to distinguish the structural similarity from conservation of native functions of these proteins.

9c. Even if BRI1 and EMS1 do naturally (not just in chimeric setting) share some downstream factors, the downstream factors should not be referred to as conserved, because "conserved" is used for different genes/proteins with the same or highly similar functions. Here the hypothesis is that the same downstream components are involved in both BRI1 and EMS1 signaling pathways.

9d. In experiments using PpEMS1 and PpTPD1, the language is again incorrect when they are said to have conserved function. First, mosses do not have anthers or pollen grains, so the PpEMS1 and PpTPD1 genes cannot possibly have similar functions to those of the Arabidopsis EMS and TPD1 genes. Second, the crucial test of conserved functions in the moss is missing, even if there is a possible conserved function.

Here again, the authors have confused structural similarity with functional conservation. The results indicate that the moss genes have sufficient sequence similarity (probably structural similarity) to substitute the genes in Arabidopsis. These results are far from sufficient to claim that the moss genes have conserved functions.

9e. The results provide some support the idea that both BRI1 and EMS1 regulate some of the same downstream components, such as BES1, which was previously identified as components of the BR signaling pathway. However, to demonstrate that BES1 (and others) are downstream of EMS1, it is necessary to show that BES1 acts downstream of EMS1 in the anther.

It is not sufficient to show that EMS1 ICD can substitute that of BRI1 to cause similar changes in BES1 (reduced phosphorylation) in the leaves.

9f. Therefore, the title of the manuscript and subtitles and titles for figures all should be modified to avoid the claims of conservation. Specifically,

In the title of the manuscript, the word "conserved" should be removed, because the conservation of the downstream components have not been shown. Also, the support for "shared with brassinosteroid receptor BRI1" is weak, as explained above.

9g. Line 164: the subtitle is not supported, because the events triggered by EMS1-TPD1 have not

been shown.

Line 165, the statement "that the ICDs of BRI1 and EMS1 have an identical function" is not supported by the results, because the rescue is not complete.

If this statement refers to native function, then it lacks supportive evidence.

10. Other writing problems:

Line 106-107, the words "the final biological output is restrained by its ICD" are not accurate. From the information presented, the final biological output is clearly also limited by the ECD, as it provides the link to the signal. In addition, the expression pattern of the gene also constrains the biological output, because all of the downstream components also need to be expressed in the same cell and at the same time. The sentence is too restrictive.

Line 137, the logic from "if EMS1 and BRI1 are recently diverged" to "their ICDs would be functionally conserved" is not clear. In addition, how recent is "recent"? If the divergence is recent compared to land plants, then the divergence can still be quite old, if it predates the separation of monocots and eudicots.

11. Figure 3c, the anther photographs are not as good as some other photos; they should be replaced with photos of higher quality.

12. The English writing of the manuscript should be further improved. The following are some examples of places that need improvements.

Lines 60, 67, 69, 71, 72, the word "will" is not necessary.

Line 70, "will be" can be changed to "becomes".

Lines 78 and 79, the words "inner stem cells" and "outer stem cells" should be changed to names already introduced in the previous paragraph. Alternatively, these terms should be introduced in the previous paragraph, providing the space and temporal context of these cells.

Line 97, "there are" should be "they are".

There are others.

Reviewer #2 (Remarks to the Author):

In work by Zheng&Bai et al authors, describe overlaps between brassinosteroid and EMS1-TPD1 signalling pathways. Authors demonstrate how intracellular domains of BRI1 and EMS1 receptors are functionally interchangeable, both in Arabidopsis and Physcomitrella and postulate origins of BRI1 receptor in angiosperms and mechanisms of RLK evolution. This work is highly complementary to the companion manuscript by Chen&Lv et al. This extremely valuable data is of great interest for the plant community research and I recommend it to be published in Nature Communication journal after some revision.

Major

When co-expressed simultaneously in bri1-116 background, EMS1 and TPD1 can partially complement mutant dwarf phenotype. Authors argue that partial complementation cannot be explained simply by expression levels of receptor and ligand. However, this could be the explanation as the ICD of EMS1 completely rescued bri1. Since BR responses are highly dose-dependent, even the small changes in signaling level could lead to different developmental outcomes, so to test how high the BR signaling level is, BES1 phosphorylation levels should be followed. To more precisely answer the question of possible full complementation of bri1-116 phenotype by ectopic expression of EMS1-TPD1 module, authors might consider synthesis of the ligand peptide and dose-dependent treatments of pBRI1:EMS1/bri1-116 lines. It is possible that EMS1/TPD1 complemented bri1 lines display more BR signaling as also shown by the observed accumulation of dephosphorylated BES1 in the blot (better shown in the companion paper of Chen & Lv et al), which is also corroborated by the observed resistance to BRZ in Fig. 2C. This can also be discussed.

This work relies on the use of mainly transgenic lines. However, the authors do not include any information about how many independent lines were characterized to record the presented in the figures phenotypes, were ever the expression levels analysed (also in relation to the comment

above).

Fig. 3 and SFig. 2h, authors do record BES1 dephosphorylation events in these transgenic lines, but the quality of the blots is not sufficient. It is very hard to conclude anything from those and I recommend that they should be repeated.

Fig. 3b, the RNAseq data require validation and the actual data should be included, alternatively not used in the paper.

SFig. 1, the authors can include some numbers for comparison of the phylogenetic relationship between the different RLKs. For example in the text many times is mentioned that EMS1 is more close to PSKR1 based on sequence comparison but what does this mean, also in relation to BRI1. Minor

Pg. 5 line 116, is stated that "bri1 null mutants display extreme dwarfism with almost normal pollen" and citations 9-13 are provided. However, none of those are related to bri1.

Pg. 7 line 196, on the blot BES1 is detected therefore the authors should not conclude that this is the same for BZR1 without testing it. Therefore this should be rephrased.

Pg. 8 line 232, there is a discrepancy between what is said in the text (line 232-233) and the Supplementary Fig. 5c. I guess the figure label is the correct version. Is proBRI1-BRI1-GFP not proBRI1-EMS-GFP.

Fig. S4 all panels will benefit from respective controls, panels (e-h) by some quantification and panel (i) by including the promoters used to complement serk1/2 double mutant as for the ems1. In the methods were genomic or cDNA sequences used to generate all constructs.

Text of the manuscript needs to be edited by a native English speaker. There are many grammatical errors and some sentences are barely understandable.

Point-by point response highlighted in blue

Reviewers' comments:

Reviewer #1 (Remarks to the Author):

Zheng et al. present results from the analysis of the EMS1 and BRI1 receptor-like protein kinases using chimeric proteins in mutant backgrounds, and other experiments. These results are suggestive of structural similarity between BRI1 and EMS1, but do not provide sufficient evidence about the molecular mechanisms of the EMS1 protein. (While we mostly agree with the reviewer's point. Unfortunately, we have not done anything about the structure of EMS1 and BRI1. Thus, it might not be best for us to deduce structural similarity between BRI1 and EMS1. In addition, we are not so much interested in studying the EMS1 protein per se. For example, we are not trying to study how EMS1 binds its ligand, how it activates its kinase activity, or what the three-D structure is, and so on. Therefore, we are sorry but we would not be able to provide sufficient evidence about the molecular mechanisms of the EMS1 protein).

The authors used "functional conservation" to refer to structural similarity of the ICDs of these proteins, as discussed in detail below. We now refrained from using the "functional conservation" throughout our manuscript. The results presented are not sufficient to provide new insights into the function of the EMS1-TPD1 signaling module. (We now used the genetic mutants and EMS1 promoter driven constructs to show that BES1 and BZR1 were functioned in the anther, specifically in tapeta, please see Fig. 3a-b). It is possible that the BRI1-EMS1 chimera functions in the transgenic plants in a way similar to the BRI1 native protein; this is supported by the partial increase of the non-phosphorylated BES1 protein (in the leaves? We now indicated that we used the 10-day old seedlings). The genetic results also support the idea that the EMS1-BRI1 chimera can function like EMS1 protein in transgenic plants, but the downstream factors for this pathway is unknown, because no experiments were performed to detected changes in downstream proteins in the anther (We now used the genetic mutants and EMS1 promoter driven constructs to show that BES1 and BZR1 were functioned in the anther, specifically in tapeta, please see Fig. 3a-b).

Therefore, the results are too preliminary to make firm conclusions regarding the mechanisms of EMS1.

(It is true that the downstream factors of this (TPD1-EMS1) signaling pathway are unknown before our study. But our study has uncovered the downstream factors of this pathway, which are BES1, BRZ1 and BIN2. We began our study by domain-swaps to see whether the kinase domains of BRI1 and EMS1 were more or less interchangeable. When we got positive results as expected, we then tested whether BRI1 and EMS1 could act in each other's native functional niche by using each other's promoter. We found that expression of BRI1 alone could partially function in tapeta (the place where EMS1 normally functions), but expression of EMS1 alone could not. We reasoned that TPD1 was not expressed or expressed at a very lower level in the BRI1's niche. We thus expressed its ligand gene TPD1 using BRI1 promoter. We now had EMS1 and TPD1 expressed together in the BRI1's functional niche. We observed good phenotypes in these transgenic plants. They largely rescued the bri1 null mutant phenotypes. With these results, we were confident that we were on the right track and proposed that EMS1-TPD1 signaling pathway and BRI1-BR signaling pathway used similar downstream factors for their signaling. Since the best known downstream factors of the BR signaling pathway are BES1, BZR1 and BIN2, we used their well-known genetic mutants bes1 and bzz1 (dominant positive mutants of the transcriptional factors of the BRI1-BR signaling pathway) to directly test whether they could rescue ems1 null mutants and its coreceptor serk1serk2 null mutants as they could rescue bri1 null mutants. Indeed, they partially rescued their fertility and almost completely rescued the tapeta! We thus identify BES1 and BZR1 as the downstream factors of this pathway (EMS1-TPD1) in tapeta! To further confirm this finding, we respectively expressed bes1 and bzz1 genes under the control of EMS1 promoter in ems1 null mutants. We showed that the expression of bes1 and bzz1 genes had similar effects as their genetic mutants had in ems1 null mutants. We further studied the negative regulator BIN2 of BR signaling pathway. We circumvented the highly redundant problem by testing if genetic mutant bin2, a dominant mutant of the negative

regulator of BR signaling pathway could suppress the phenotypes of the co-overexpression lines of *EMS1* and *TPD1* under *BRI1* promoter. Indeed it could suppress the phenotypes of the co-overexpression lines of *EMS1* and *TPD1*. Altogether, our study undoubtedly establishes that *BES1*, *BZR1* (in the anther, in fact in the tapeta) and *BIN2* or its homologues are the downstream factors of *EMS1-TPD1* signaling pathway for the first time. As such, our results presented in the manuscript are now able to provide new insights into the function of the *EMS1-TPD1* signaling module. As you can see, we are not drawing our conclusion solely based on the domain-swap of *BRI1* and *EMS1*).

The following specific comments must also be addressed before the manuscript can be published.

1. For the transgenic plants shown in Figure 1, information on the number of transgenic lines generated for each construct should be provided; in addition, the expression levels of the transgene, and the corresponding phenotypes for at least two independent transgenic lines should be presented, to avoid the possibility that some of the phenotypes and other properties of the transgenic plants might be caused by unknown mutations generated during the transformation process.

We now stated in the methods that at least 20 transgenic lines were obtained for each construct. We also provided protein expression levels in the Fig. 1d and 1g. Additional three independent transgenic lines for the constructs key to our conclusion were listed in Supplementary Fig. 2, for phenotypes (2a and 2c) and for protein expression (2b and 2d). We also re-did the Western blot for the dephosphorylation of *BES1*, a hallmark of active BR signaling pathway, thus providing a better evidence for the functional similarity of the chimeric receptor (*BRI1^{ECD}::EMS1^{ICD}*) and the *BRI1* (*ECD*, Extracellular Domain and *ICD*: intracellular Domain).

2. In the description of the phenotypes of the transgenic lines (from perhaps a single line here), some differences between *BRI1* and *BRI1-EMS* have been overlooked. For example, the hypocotyl lengths of the *BRI1-EMS* seedlings in the absence of *BRZ* are less than those of the transgenic *BRI1* seedlings. The description should be accurate and truly reflect the results.

We now added "partially" in the figure legend and main text.

3. In addition, in Figure 2, the hypocotyl lengths of seedlings expressing both *EMS1* and *TPD1* are intermediate between *Col-0/BRI1* transgenic and the *bri1* mutant, indicating that the *EMS1/TPD1* transgenes can only partially rescue the mutant phenotypes. Again the description should be revised.

We agree with the reviewer. We thus added "partially rescue or similar expression" in the figure legend and main text

4. Also, in Figure 2, although *BRI1* was able to promote the formation of four layers of somatic cells, the *EMS1* transgenic anther shows more cell layers in comparison. Furthermore, the fertility of the *BRI1* plants is less than that of the *EMS1* plants. Therefore, the ability of the *EMS1* and *BRI1* genes to rescue the *ems1* mutant is somewhat different. The description should be consistent with this fact.

EMS1 and *BRI1* genes might be diverged 400 millions year ago. Each receptor would have accumulated functions that differentiate from each other the course of evolution. We agree that the functions of *EMS1* and *BRI1* are not identical, thus we accept the reviewer's suggestion to be mindful on our wording and made needed corrections throughout the main text and figure legends.

5. In Figure 3, the tissues used for western experiments should be explicitly stated. Were leaves? Were they flowers? The phosphorylation state of *BES1* protein in the plants with the *EMS1* and *TPD1* transgenes is intermediate between those of the *BRI1* transgenic plants with or without *BL* added. These results suggest that although *EMS1 ICD* has a similar function to that of *BRI1*, but it is weaker.

We now clearly stated that we used 10-day old seedlings for the Western blot of *BES1*. We repeated *BES1* dephosphorylation experiment and provided better Western blot and showed in new Fig. 2d (to replace original Fig. 3a) and Fig. 1d (to replace original supplementary Fig. 2h). The results suggest that although *EMS1 ICD* has a similar function to that of *BRI1*, but it is weaker.

6. Also, in Figure 3, the transcriptome analyses clearly showed that the *EMS1-TPD1* transgenic plants have distinct sets of both up-regulated and down-regulated genes. Thus these genes do not act like the *BRI1* gene. This should be clear.

We agree with the concern that the reviewer has. Per requested by the other reviewer, we deleted the original Fig. 3b and added verification by Q-PCR shown in Supplementary Fig. 5c-e. We agree with the reviewer that EMS1-TPD1 signaling system is not identical to BRI1-BR signaling system. They just use some of the same downstream components for their signaling pathways without accounting for their amplitudes and additional possible specific components of each system.

7. The phylogenetic tree in Figure 4f is inaccurate, because lycophytes are not monophyletic (do not form a clade or branch); ferns are sister to seed plants and lycopods form a separate branch (Lycophytes form two branches). The situation for bryophytes is less clear. Some studies support monophyly of bryophytes, but others do not (with two or three branches). Also, there are several groups (branches) within charophytes, and some are more closely related to land plants than others. Therefore, charophytes are also not monophyletic.

The reviewer is completely right! We now deleted lycophytes from the original Fig. 4f and replaced it by Fig. 4h. Since our focus is to give the reader some idea of the origin of BRI1 and EMS1, rather than in depth study of the evolution of BRI1 and EMS1. We hope that the reviewer can accept us to keep bryophytes and charophytes to form a very rough monophyletic tree of plants for now.

8. The phylogenetic analyses shown here in Supplemental Figures 1 and 6 should be described with more details, including the parameters for sequence retrieval, for alignment, and bootstrap values should be obtained and shown on the trees.

We now provided more details in the figure legends, and replaced the tree with bootstrap values in Supplementary Fig. 1 and replaced Supplementary Fig. 6 by supplementary Fig. 8 with bootstrap values.

9. The writing of the manuscript contains a number of misconceptions and misstatements about interpretation of results and evolutionary conservation.

We have modified our wording throughout the manuscript.

For example (there are more, sometimes similar situations):

9a. The use of transgene (including chimeric genes) in a mutant background can generate situations that suggest one gene might be able to act to substitute another gene. For example, the ICD of EMS1 could substitute that of BRI1 in transgenic plants. However, these results do not mean that the two genes being tested have evolutionarily conserved functions. The words “evolutionarily conserved functions” refer to their functions in the wild-type plants of the native species, not in transgenic plants generated in the lab (even in a heterologous plant).

In fact, these two genes, EMS1 and BRI1, clearly have distinct native functions as indicated by their dramatically different mutant phenotypes. So, they should not be described as having “conserved functions”.

We have adjusted our wording throughout the manuscript.

9b. In addition, rescue (partial rescue) of a mutant phenotype does not necessarily mean that the mutated gene and the rescuing gene are evolutionarily conserved. In fact, sometimes, a downstream gene can rescue a mutant in the upstream gene, when the two genes encode proteins of dissimilar sequences. For example, here BRI1 and BRI-EMS1 fusion were shown to partially rescue a *det2* mutant, but BRI1 and EMS1 clearly do not have conserved functions with respect to DET2.

The correct interpretation is that the EMS1 ICD has sufficient sequence and (probably) structural similarity to that of BRI1 to be able to partially substitute for the BRI1 when the EMS1 ICD is fused to the ECD of BRI1. These and other results in the manuscript do not indicate evolutionarily conserved functions, only structural similarity. The structural similarity would allow the chimeric protein to interact with some of the downstream factors of BRI1, but this is not sufficient to indicate that EMS1 normally interacts with such downstream factors.

The writing related to such experiments need to distinguish the structural similarity from conservation of native functions of these proteins.

We refrained from using “evolutionarily conserved functions”.

9c. Even if BRI1 and EMS1 do naturally (not just in chimeric setting) share some downstream factors (we now do show that BES1 and BZR1 NATURALLY acts downstream of EMS1 in the anther, in fact, in the tapeta by suppression of *ems1* mutants by their

native genetic mutants *bes1D* and *bzr1D*), the downstream factors should not be referred to as conserved, because “conserved” is used for different genes/proteins with the same or highly similar functions. Here the hypothesis is that the same downstream components are involved in both BRI1 and EMS1 signaling pathways.

I agree with what the reviewer said (“conserved” is used for different genes/proteins with the same or highly similar functions). In our case, EMS1 and BRI1 are in a family tree sister to each other; their sequences are similar (they share high identity); their intracellular domains are exchangeable; they can produce similar function if they are in the same place; they even have the same target genes (BES1 and BZR1); they have the same group of coreceptors; they might have the same group of negative regulators (BIN2 family).

9d. In experiments using PpEMS1 and PpTPD1, the language is again incorrect when they are said to have conserved function. First, mosses do not have anthers or pollen grains, so the PpEMS1 and PpTPD1 genes cannot possibly have similar functions to those of the Arabidopsis EMS and TPD1 genes. Second, the crucial test of conserved functions in the moss is missing, even if there is a possible conserved function.

Here again, the authors have confused structural similarity with functional conservation. The results indicate that the moss genes have sufficient sequence similarity (probably structural similarity) to substitute the genes in Arabidopsis. These results are far from sufficient to claim that the moss genes have conserved functions.

We modified our wording and refrained from using “conserved functions”.

9e. The results provide some support the idea that both BRI1 and EMS1 regulate some of the same downstream components, such BES1, which was previously identified as components of the BR signaling pathway. However, to demonstrate that BES1 (and others) are downstream of EMS1, it is necessary to show that BES1 acts downstream of EMS1 in the anther. (This argument of the reviewer is right but we now have provided that BES1 and BZR1 acts downstream of EMS1 in the anther, in fact, in the tapeta by their native genetic mutants *bes1D* and *bzr1D*).

It is not sufficient to show that EMS1 ICD can substitute that of BRI1 to cause similar changes in BES1 (reduced phosphorylation) in the leaves.

We agree with the reviewer, but we are not drawing our conclusion solely based on this experiment. Rather, we have conclusive genetic evidence to support our conclusion.

9f. Therefore, the title of the manuscript and subtitles and titles for figures all should be modified to avoid the claims of conservation. Specifically,

In the title of the manuscript, the word “conserved” should be removed, because the conservation of the downstream components have not been shown [we now have shown that EMS1 and BRI1 had the same downstream components, BES1 and BZR1 (undoubtedly) and BIN2 (likely)]. Also, the support for “shared with brassinosteroid receptor BRI1” is weak, as explained above.

We removed the “conserved” from the title as requested.

9g. Line 164: the subtitle is not supported, because the events triggered by EMS1-TPD1 have not been shown.

Line 165, the statement “that the ICDs of BRI1 and EMS1 have an identical function” is not supported by the results, because the rescue is not complete.

If this statement refers to native function, then it lacks supportive evidence.

We modified these statements as requested.

10. Other writing problems:

Line 106-107, the words “the final biological output is restrained by its ICD” are not accurate. From the information presented, the final biological output is clearly also limited by the ECD, as it provides the link to the signal. In addition, the expression pattern of the gene also constrains the biological output, because all of the downstream components also need to be expressed in the same cell and at the same time. The sentence is too restrictive.

We changed it to “the final biological output is produced through its ICD (or KD)”.

Line 137, the logic from “if EMS1 and BRI1 are recently diverged” to “their ICDs would be functionally conserved” is not clear. In addition, how recent is “recent” ? If the divergence is recent compared to land plants, then the divergence can still be quite old, if it predates the separation of monocots and eudicots.

We rephrased this content (if EMS1 and BRI1 were sister groups, their ICDs might be functionally closer than their ECDs) and showed in line 134-135.

11. Figure 3c, the anther photographs are not as good as some other photos; they should be replaced with photos of higher quality.

We now replaced the Fig. 3c with Fig. 3a with photos of higher quality.

12. The English writing of the manuscript should be further improved. The following are some examples of places that need improvements.

Lines 60, 67, 69, 71, 72, the word “will” is not necessary.

Line 70, “will be” can be changed to “becomes” .

We have made the changes as requested

Lines 78 and 79, the words “inner stem cells” and “outer stem cells” should be changed to names already introduced in the previous paragraph. Alternatively, these terms should be introduced in the previous paragraph, providing the space and temporal context of these cells.

We made the change as the reviewer suggested.

Line 97, “there are” should be “they are” .

We made the change as the reviewer suggested.

There are others.

We revised the whole manuscript.

Reviewer #2 (Remarks to the Author):

In work by Zheng&Bai et al authors, describe overlaps between brassinosteroid and EMS1-TPD1 signalling pathways. Authors demonstrate how intracellular domains of BRI1 and EMS1 receptors are functionally interchangeable, both in Arabidopsis and Physcomitrella and postulate origins of BRI1 receptor in angiosperms and mechanisms of RLK evolution. This work is highly complementary to the companion manuscript by Chen&Lv et al. This extremely valuable data is of great interest for the plant community research and I recommend it to be published in Nature Communication journal after some revision. (We appreciate the reviewer’s comments)

Major

When co-expressed simultaneously in bri1-116 background, EMS1 and TPD1 can partially complement mutant dwarf phenotype. Authors argue that partial complementation cannot be explained simply by expression levels of receptor and ligand. However, this could be the explanation as the ICD of EMS1 completely rescued bri1. Since BR responses are highly dose-dependent, even the small changes in signaling level could lead to different developmental outcomes, so to test how high the BR signaling level is, BES1 phosphorylation levels should be followed. To more precisely answer the question of possible full complementation of bri1-116 phenotype by ectopic expression of EMS1-TPD1 module, authors might consider synthesis of the ligand peptide and dose-dependent treatments of pBRI1:EMS1/bri1-116 lines. It is possible that EMS1/TPD1 complemented bri1 lines display more BR signaling as also shown by the observed accumulation of dephosphorylated BES1 in the blot (better shown in the companion paper of Chen & Lv et al), which is also corroborated by the observed resistance to BRZ in Fig. 2C. This can also be discussed.

We repeated BES1 dephosphorylation experiment and provided better Western blot shown in new Fig. 2d (to replace original Fig. 3a) and Fig. 1d (to replace original Supplementary Fig. 2h). Because TPD1 is a 20KD protein, it is difficult to synthesize the ligand and even we can synthesize the ligand, it might not be able to intake into the whole plants unless we do biochemical

study with protoplasts, which might not be more reliable. We thus selected a transgenic line of *proBRI1::EMS1* to cross to three independent transgenic lines of *proBRI1::TPD1* with different levels of *TPD1* expression. We did observe a good correlation between the expression levels of the ligand and the degree of the complementation of *bri1-116* phenotypes.

This work relies on the use of mainly transgenic lines. However, the authors do not include any information about how many independent lines were characterized to record the presented in the figures phenotypes, were ever the expression levels analysed (also in relation to the comment above).

We now added the numbers of transgenic lines obtained in our study in the methods, and provided protein expression levels for all transgenic lines that we selected to use in our manuscript. When the antibodies were not available (in the case of the expression of *TPD1*), semi-quantitative PCR was used to determine the transcription.

Fig. 3 and SFig. 2h, authors do record BES1 dephosphorylation events in these transgenic lines, but the quality of the blots is not sufficient. It is very hard do conclude anything from those and I recommend that they should be repeated.

We now repeated BES1 dephosphorylation experiment and provided better Western blot and showed in new Fig. 2d (to replace original Fig. 3a) and Fig. 1d (to replace original supplementary Fig. 2h).

Fig. 3b, the RNAseq data require validation and the actual data should be included, alternatively not used in the paper.

We now deleted the original Fig. 3b and added verification by Q-PCR, which was shown in Supplementary Fig. 5c-e.

SFig. 1, the authors can include some numbers for comparison of the phylogenetic relationship between the different RLKs. For, example in the text many times is mentioned that EMS1 is more close to PSKR1 based on sequence comparison but what does this mean, also in relation to BRI1.

We now replaced the tree with bootstrap values, which indicates EMS1 and PSKR1 belong to a branch that is sister to BRI1 and BRLs (BRI1-likes).

Minor

Pg. 5 line 116, is stated that “*bri1* null mutants display extreme dwarfism with almost normal pollen” and citations 9-13 are provided. However, none of those are related to *bri1*.

We are apologized for the careless mistake and now we quoted the right references.

Pg. 7 line 196, on the blot BES1 is detected therefore the authors should not conclude that this is the same for BZR1 without testing it. Therefore this should be rephrased.

We removed BZR1 from the original line 196 and showed in the new line 204.

Pg. 8 line 232, there is a discrepancy between what is said in the text (line 232-233) and the Supplementary Fig. 5c. I guess the figure label is the correct version. Is *proBRI1-BRI1-GFP* not *proBRI1-EMS-GFP*.

We made a mistake in old Supplementary Fig. 5c, we now changed “*proBRI1-BRI1-GFP*” to “*proBRI1-EMS1-GFP*” and showed it in new supplementary Fig. 7d.

Fig. S4 all panels will benefit from respective controls, panels (e-h) by some quantification and panel (i) by including the promoters used to complement *serk1/2* double mutant as for the *ems1*.

For original supplementary Fig. 4e-h (new supplementary Fig. 6a-d), we now added typical examples of siliques with fertilized seeds in new supplementary Fig. 6e-f, images of siliques with seeds in supplementary Fig. 6e and quantification of fertilized seeds in the siliques in supplementary Fig. 6f. For original supplementary Fig. 4i (new supplementary Fig. 6g), except the ones that we labeled with *pro^{EMS1}::bzi1-1D* and *pro^{EMS1}::bes1-D* are transgenes, the other are genetic mutants, thus we are reluctant to add promoters. We also added the images that indicated the reappearance of the tapeta in the developing anthers of triple mutants (*bzi1-D serk1/2* and *bes1-D serk1/2*) and transgenic plants of *pro^{EMS1}::bzi1-1D* and *pro^{EMS1}::bes1-D* in *ems1* background.

In the methods were genomic or cDNA sequences used to generate all constructs.

We now added the information to the methods.

Text of the manuscript needs to be edited by a native English speaker. There are many grammatical errors and some sentences are barely understandable.

We now carefully edited the manuscript.

Reviewers' comments:

Reviewer #1 (Remarks to the Author):

Zheng and Bai et al. have addressed some of the comments and concerns raised on the previous manuscript, especially regarding conservation between BRI1 and EMS1. However, there remain a number of important problems, as indicated below.

Major comments

1. Prior to this study, it is known that BRI1 and EMS1 have different and important functions. In addition, BRI1 and EMS1 are similar in protein sequences, particularly in the kinase domain, more than other RPKs.

The authors have shown that the BRI1 and EMS1 proteins can partially rescue the mutant of the other gene, if expressed under the promoter of the other gene. These experiments further support the idea that these two proteins are similar, and when expressed in the right cells, can interact with components of the signaling pathway triggered normally by the other protein.

These experiments per se do not provide new insights about the mechanisms of these proteins, particularly the nature of downstream components for EMS1 (such information is already known for BRI1).

2. Specifically, the authors used transgenic plants and other materials to study the functions of BRI1-BR and EMS1-TPD1 signaling pathways, but have used the word "function" in an unclear way, because "function" can be used in at least two different contexts:

(1) biological (such as developmental, physiological) functions at the whole plant level; this requires the use of plants of various genotypes to test the roles of different genes. However, not all plant phenotypes can contribute to the understanding of biological function significantly. For example, expressing EMS1 under the BRI1 promoter and showing the transgene can substitute the BRI1 gene does not add much new information about EMS1 biological function, which was previously tested using the *ems1* mutant and other methods.

(2) molecular (biochemical) functions; this often needs molecular interactions with specific genes or proteins. Sometimes inferences can be made about molecular functions using molecular genetic experiments, but these inferences are weak without direct evidence from molecular/biochemical experiments.

The authors need to be more clear which specific meaning they are using in different places of the paper.

3. For the use of "function", it is not clear sometimes what the authors meant when they discuss the "functions" of BRI1 and EMS1. As mentioned, the BRI1 and EMS1 genes clearly have very different biological functions, and the authors recognize this. However, the authors claim that these proteins have the same or similar functions (for example, the subtitle on line 186), primarily based on the results chimeric genes (promoter swapping). Here, the implied function should be molecular/biochemical functions. In other words, the use of "function" without clear specification about the meaning of the word makes the claims unclear.

4. Third, the above results support the idea that the activated protein kinase domain of EMS1 can replace the activated BRI1 protein kinase domain in the BRI1 developmental domain (for vegetative development). However, how EMS1 does this is not known from the phenotypes. Because the rescue was partial, other explanations could not be eliminated, as ectopic expression of genes can cause artifacts that are not related to native functions.

5. Because the EMS1-TPD1 signaling module was able to partially suppress the *bri1* mutant phenotypes when both EMS1 and TPD1 were co-expressed from the BRI1 promoter, it is expected that normal downstream components of BRI1 signaling pathway would also behave in ways similar in the proBRI1-EMS1 proBRI1-TPD1 transgenic plants (such as the dephosphorylation of BES1). Otherwise the phenotypes of these transgenic plants would be hard to explain. These results can

further support the idea stated above in comment 3, but again the transgenic plants can also yield artifacts.

6. In other words, the experiments mentioned above in comments 3 and 4 together provided some support the idea that EMS1-TPD1 can replace BRI1 function in vegetative development, possibly via mechanisms similar to that of BRI1. However, the last point is not strongly supported, because it was NOT shown that the dephosphorylation of BES1 in the proBRI1-EMS1 proBRI1-TPD1 transgenic plants was important for the observed phenotypes and it is NOT known how EMS1 regulated the dephosphorylation of BES1.

7. The authors also expressed BRI1 under the EMS1 promoter, and found partial rescue of the *ems1* sterility phenotype. Again, these experiments showed that the EMS1 and BRI1 proteins are similar and can possibly behave similarly at the molecular level, although there is no direct mechanistic evidence. The same concerns about the use of transgene to draw conclusions about function remains, without additional support.

8. The experiments of using gain-of-function alleles of BES1 and BZR1 to partially rescue the *ems1* mutant do provide support for the idea that BES1/BZR1 might act downstream of EMS1. However, gain-of-function results should be compared with loss-of-function results, which provide more definitive support. In other words, results from BES1 and BZR1 mutants are necessary to make the claim that BES1 and BZR1 are important for male fertility.

9. Also, loss-of-function of BIN2 should also be tested for effects on fertility in an EMS1-dependent pathway.

10. Another important experiment is to show that BES1 and BZR1 are not dephosphorylated in the *ems1* mutant flowers, similar to the experiments shown for the *bri1* mutant.

11. The experiments using homologs from *Physcomitrella* support the idea that these homologous proteins are similar to the Arabidopsis EMS1 and TPD1 protein to substitute EMS1 and TPD1 in Arabidopsis.

These experiments do not show that the *Physcomitrella* proteins can cause similar downstream changes in *Physcomitrella*; these proteins certainly do not support pollen development in Bryophytes because Bryophytes do not produce pollen.

Therefore, the experiments do not support the claim that EMS1 and TPD1 have conserved functions (biological or molecular) functions in land plants.

Because BRI1 can also rescue the *ems1* mutant when BRI1 is expressed from the EMS1 promoter, the experiments with PpEMS1 does not inform regarding the native function of PpEMS1.

12. To investigate the conservation of EMS1 in land plants, experiments to test the function of PpEMS1 in *Physcomitrella* must be performed, including expression studies and molecular genetic tests. Therefore, the conclusion stated in the subtitle on lines 277-288 is not supported by the results.

In summary, the manuscript contains largely confirmatory results and the results do not provide sufficient support for several conclusions. The overall manuscript does not lead to sufficient new insights regarding EMS1 function and related genes to warrant publication in *Nature Comm*. The manuscript in its current form is also not suitable in a more specialized journal, because the authors have not demonstrated a clearly understanding of the implications of the results, and have not presented the results properly.

Minor comments: there are numerous writing problems. The following are some examples.

1. Line 57, the sentence should be change to "apical meristem into the inflorescence and floral meristems by a combinative action of flowering transition and floral meristem genes". In addition,

the ref 7 is not a correct reference.

2. Even for floral organ identity genes, ref 7 is for only one type of floral organ identity genes (E-function), but there are also other floral organ identity genes (ABC).
3. Line 64, again, ref 7 is not a correct reference for floral meristem.
4. Line 110, the sentence is missing a word at the end, such as "domain" or "segment".
5. Line 129, it is not clear what the authors meant by "common function". Clearly these two genes affect different aspects of plant development. It is true that BRI1 and EMS1 are phylogenetically close compared with many other RLKs. However, many phylogenetically close genes have different biological functions. BRI1 and EMS1 also have distinct biological functions.
6. Line 131, the sentence is incomplete. BRI1 and EMS1?
7. Line 134, authors have indicated that BRI1 and EMS1 are known to share coreceptors, so it is not clear why they claim to have revealed this.
8. There are other places where the statements about results or conclusions are inaccurate.

Reviewer #2 (Remarks to the Author):

The authors have addressed most of the concerns and I am convinced that the BRI1 and EMS1 pathways are sharing signaling components. I still think this work is of a high interest and it should be published in Nature Communications. I only have a few requests for a minor revision before publication:

- The co-expression of TPD1 and EMS1 partially complements *bri1-116* mutant phenotype. Actually, the plants show signs of increased levels of BR signaling based on the phosphorylation status of BES1, which is in state of constitutive ON state (Fig 2 d). Increased levels of activated BES1 can be explained by high levels of the ligand in the transgenic plants. The authors argue that TPD1 cannot be chemically synthesized and exogenously applied and they provide an elegant solution by crossing *proBRI1:EMS1* with three different *proBRI1:TPD1* transgenic lines with different TPD1 expression levels (FIG S4 a-b). However, the line (# 4) with higher TPD1 expression levels is phenotypically closer to wild type plants, while BES1 phosphorylation status is closer to wild type in line #12, which has the lowest TPD1 expression level. Additionally, the expression of CPD gene is slightly upregulated and BAS1 downregulated in transgenic *EMS1&TPD1/bri1-116* plants when compared to *BRI1/bri1-116* (Fig S5 c,e). The author should indicate which transgenic lines were used for qPCR, #4, #15 or #12? This result suggests that the BES1 activated by EMS1-TPD1 signaling pair, cannot perform exactly the same functions of BRI1-BR activated BES1 during maintenance BR homeostasis, since expression of these two genes shows opposite trend in increased BR signaling conditions. This might imply that BES1 transcription factor, although common for the two signaling pathways, is somehow differently controlled within the two pathways via perhaps, differential phosphorylation, interaction partners, etc? The authors should discuss this possibility in the discussion section?
- Authors included more information on the number of transgenic lines, stating that they used at least 20 independent transgenic lines. How many of those were analyzed per construct?
- BES1 blots have been improved and have sufficient quality for publication.
- The biggest issue for this manuscript is the quality of written English. The style of writing is still far from acceptable, in fact, has gotten even worse. As suggested during the first round of revision, authors need to send the text to an academic proofreader. Here are the examples of unsatisfactory language:
 - Line 51, Avoid repeating terms within the sentence (cell surface in this case)
 - Line 55, Plant life cycle starts with saprophytic seed germination, not plant life.
 - Line 61, Do not start a sentence with "At the end". Better is Finally.
 - Line 110, followed by transmembrane "domain".
 - Line 122, "Activated BES1/BZR1 then negatively regulates BR biosynthesis but positively controls BR metabolism to control the hormonal homeostasis." Did authors mean catabolism instead of metabolism?
 - Line 146, "ligand-binding domains are evolved" – rephrase this.

Line 148, EMS1 and BRI1 are not sister groups, they might belong to sister groups.

Line 166, "mutants were indistinguishable by eyes" – rephrase.

Line 167, "Furthermore, among all the reliable molecular markers, including sensitive to BRs and BR inhibitors, dephosphorylation of the transcriptional factor BES1/BZR1, downregulation of the expression of the BR biosynthetic genes and upregulation of the expression of the BR metabolic genes, used to judge the active BR response, only the BR metabolic gene BAS1 showed a lesser effect." This sentence is a typical example of poor writing style. Again, professional proofreading is necessary!

Line 186, "Signaling pairs of EMS1-TPD1 and BRI1-BRs trigger similar functions." Title of this paragraph should be changed to Signaling pairs of EMS1-TPD1 and BRI1-BRs trigger similar molecular responses.

Line 193, erase together from this sentence

Line 224, "BES1 was accumulated" should stand BES1 is dephosphorylated and accumulates...

Line 262, " ... why do plants have both receptors?" Avoid layman's terms and fatalistic statements/questions.

Line 279, 327, 349, similar comment to line 262

Line 358, 391 Do not start sentence with Doing so.

Line 366, "Therefore, it will be best that EMS1 perceives TPD1 in tapeta while BRI1 binds BRs elsewhere to complementarily control the whole plant development through the same essential signaling pathway." Rephrase, there is no such a thing as a best solution for the plant.

Line 372, "The cell surface receptors perceive and transmit various signals from the outside of the cell and transmit into the inside of the cell to control a wide range of physiological and developmental processes." Another sentence that needs to be rephrased and written in a scientific manner.

Line 404, "like nothing else ever known." Remove this statement.

Please find the point-by-point response highlighted in blue

Reviewers' comments:

Reviewer #1 (Remarks to the Author):

Zheng and Bai et al. have addressed some of the comments and concerns raised on the previous manuscript, especially regarding conservation between BRI1 and EMS1. However, there remain a number of important problems, as indicated below.

Major comments

1. Prior to this study, it is known that BRI1 and EMS1 have different and important functions. In addition, BRI1 and EMS1 are similar in protein sequences, particularly in the kinase domain, more than other RPKs.

The authors have shown that the BRI1 and EMS1 proteins can partially rescue the mutant of the other gene, if expressed under the promoter of the other gene. These experiments further support the idea that these two proteins are similar, and when expressed in the right cells, can interact with components of the signaling pathway triggered normally by the other protein.

These experiments per se do not provide new insights about the mechanisms of these proteins, particularly the nature of downstream components for EMS1 (such information is already known for BRI1).

Because most of the questions of reviewer #1 are concerning the rationale of our approaches and the interpretation of our results, we try to address most of them altogether in the question #1:

The ultimate goal of this study was to identify EMS1-TPD1 downstream signaling components in the tapeta. For convenience in our description, we used BES1, a BRI1-specific transcriptional factor, to represent our target signaling components. Thus, the goal of the experiments was to establish if EMS1 could activate BES1. We did it indirectly through expressing *EMS1 & TPD1* or *BRI1:EMS1* in *bri1* mutants. For these experiments, we were not concerned about whether they worked exactly the same as endogenous EMS1 since we planned to show that in the next step. What we really needed to know was whether EMS1 could, and other RLKs could not activate BES1 to show its specificity. If we obtained positive results, then we would move to the next step! In this case, whether they were strong or weak, molecular or biological, ectopic or accumulated is not important since it would not affect our interpretation that EMS1 can or cannot activate BES1. As long as we found that EMS1 could activate BES1, that would be a good enough result. However, *BRI1:EMS1-KD* (kinase domain) completely rescued *bri1* phenotypes to an extent indistinguishable from WT, while *EMS1 & TPD1* partially rescued *bri1* phenotypes. This is likely due to the distinctive ligands or extracellular domains rather than the difference of the kinase functions (EMS1 and BRI1). Yet, in both cases, BES1 was dephosphorylated, confirming that the KD of EMS1 activates BES1. Based on this finding we were ready to investigate whether BES1 functions in the tapeta. However, what we initially needed to do was to show: Firstly, whether there is BES1 in the tapeta? Secondly, whether the BES1 can be activated in the tapeta? Thirdly, whether activated BES1 has a function in the tapeta? Finally, whether EMS1 activates BES1 in the tapeta?

Since BES1 is known to be required for BRI1's function, we expressed BRI1 that is not expressed in tapeta in *ems1* mutants. We observed a complete and partial recovery of the *ems1* phenotypes by *EMS1:BRI1* and *BRI1*, respectively. These results imply that there is native BES1 that can be activated in the tapeta. To further address whether BES1/BZR1 can control tapetal development, we used *bes1D/bzr1D* dominant mutant (with more phosphorylated BES1 and dephosphorylated BES1) to biologically complement the *ems1* phenotypes. Indeed, they could almost completely rescue *ems1* mutants. Since native *BES1* expression was ubiquitous, there was a chance that this could be an indirect effect. We thus expressed *bes1D* in the tapeta using *EMS1* promoter to show a similar result as that for *bes1D/bzr1D* genetic mutants. Together, this implies that BES1/BZR1 controls tapetal development and that activated BES1/BZR1 alone is sufficient for controlling tapetal development. Ideally, we need to demonstrate whether BES1 is directly activated by EMS1 in the tapeta, but that experiment was too technically challenging. An alternative would be to examine the phosphorylated status of BES1 in the WT and *ems1* mutants tapeta (question 10). However, this experiment is undoable since the *ems1* mutant has no tapeta. Another option would be to treat the anthers with or without TPD1, yet TPD1 is 20KD and the tapeta are several meristematic cells down from the outermost layer. Therefore, the lack of a doable experiment makes it presently an almost impossible task. So, as we mentioned above, we used transgenic plants to demonstrate that EMS1 can activate BES1 in vivo in the BRI1 functional domain. Importantly, chimeric BRI1-EMS1 behaves completely as WT BRI1 by showing similar phenotypical, physiological, molecular and biochemical response, indirectly demonstrating that the KD of EMS1 is activated as the KD of BRI1. Since we have shown that EMS1 was expressed in the tapeta and BRI1 was not, and both EMS1 and BES1 families (from Wenqiang Tang's group and Jia's group) were required for the tapeta. Altogether, BES1 must be activated by EMS1 in the tapeta, therefore, we could conclude that native EMS1 activates native BES1 in the tapeta. As you can see, we have included steps to avoid any conclusion based only on a single experiment, and even if there are a few artifacts, it does not affect our final conclusions.

2. Specifically, the authors used transgenic plants and other materials to study the functions of BRI1-BR and EMS1-TPD1 signaling pathways, but have used the word "function" in an unclear way, because "function" can be used in at least two different contexts:

(1) biological (such as developmental, physiological) functions at the whole plant level; this requires the use of plants of various genotypes to test the roles of different genes. However, not all plant phenotypes can contribute to the understanding of biological function significantly. For example, expressing EMS1 under the BRI1 promoter and showing the transgene can substitute the BRI1 gene does not add much new information about EMS1 biological function, which was previously tested using the *ems1* mutant and other methods.

We agree with the reviewer and have made changes accordingly in the text.

(2) molecular (biochemical) functions; this often needs molecular interactions with specific genes or proteins. Sometimes inferences can be made about molecular functions using molecular genetic experiments, but these inferences are weak without direct evidence from molecular/biochemical experiments.

The authors need to be more clear which specific meaning they are using in different places of the paper.

We agree with the reviewer and have made changes accordingly in the text.

3. For the use of "function", it is not clear sometimes what the authors meant when they discuss

the “functions” of BRI1 and EMS1. As mentioned, the BRI1 and EMS1 genes clearly have very different biological functions, and the authors recognize this. However, the authors claim that these proteins have the same or similar functions (for example, the subtitle on line 186), primarily based on the results chimeric genes (promoter swapping). Here, the implied function should be molecular/biochemical functions. In other words, the use of “function” without clear specification about the meaning of the word makes the claims unclear.

We agree with the reviewer and have made changes accordingly in the text.

4. Third, the above results support the idea that the activated protein kinase domain of EMS1 can replace the activated BRI1 protein kinase domain in the BRI1 developmental domain (for vegetative development). However, how EMS1 does this is not known from the phenotypes. Because the rescue was partial, other explanations could not be eliminated, as ectopic expression of genes can cause artifacts that are not related to native functions.

We agree with the reviewer and have made changes accordingly in the text.

5. Because the EMS1-TPD1 signaling module was able to partially suppress the *bri1* mutant phenotypes when both EMS1 and TPD1 were co-expressed from the BRI1 promoter, it is expected that normal downstream components of BRI1 signaling pathway would also behave in ways similar in the proBRI1-EMS1 proBRI1-TPD1 transgenic plants (such as the dephosphorylation of BES1). Otherwise the phenotypes of these transgenic plants would be hard to explain. These results can further support the idea stated above in comment 3, but again the transgenic plants can also yield artifacts.

We agree with the reviewer. But this experiment is only intended to show whether EMS1-TPD1 CAN activate BES1 family (molecular phenotypes). This is an important question and problem, and we will study this in the next project.

6. In other words, the experiments mentioned above in comments 3 and 4 together provided some support the idea that EMS1-TPD1 can replace BRI1 function in vegetative development, possibly via mechanisms similar to that of BRI1. However, the last point is not strongly supported, because it was NOT shown that the dephosphorylation of BES1 in the proBRI1-EMS1 proBRI1-TPD1 transgenic plants was important for the observed phenotypes and it is NOT known how EMS1 regulated the dephosphorylation of BES1.

We agree with the second point but disagree with the first point. Since using *bri1* mutants that had little accumulation of dephosphorylation of BES1, but in the proBRI1-EMS1 proBRI1-TPD1 transgenic plants, the dephosphorylation of BES1 is elevated. Dephosphorylation of BES1 in the proBRI1-EMS1 proBRI1-TPD1 transgenic plants is related to the partial recovery phenotypes although their relationship is not linear (dephosphorylated BES1 in transgenics but not *bri1*). One way to the second question is to treat the transgenic with or without TPD1, yet TPD1 is 20KD but how it is processed is unclear. Thus, this remains an almost impossible task. Importantly, BRI1:EMS1-KD (kinase domain) completely rescued *bri1* phenotypes to an extent indistinguishable from WT (including BR response). Therefore, we have no reason to believe that the KD of EMS1 regulates the dephosphorylation of BES1 different from that of the KD of BRI1, although we have not directly shown.

7. The authors also expressed BRI1 under the EMS1 promoter, and found partial rescue of the *ems1* sterility phenotype. Again, these experiments showed that the EMS1 and BRI1 proteins are similar and can possibly behave similarly at the molecular level, although there is no direct mechanistic evidence. The same concerns about the use of transgene to draw conclusions about

function remains, without additional support.

Please see answer to the question #1

8. The experiments of using gain-of-function alleles of BES1 and BZR1 to partially rescue the *ems1* mutant do provide support for the idea that BES1/BZR1 might act downstream of EMS1. However, gain-of-function results should be compared with loss-of-function results, which provide more definitive support. In other words, results from BES1 and BZR1 mutants are necessary to make the claim that BES1 and BZR1 are important for male fertility.

This is an important issue but the results have been published by Wenqiang Tang's group (2019. Mol. Plants) and also by the accompanying manuscript submitted by Jia Li's group. This is why we submitted our papers together since we see these two works are complementary. We would appreciate if they are considered together. However, our paper stands alone and is self-sufficient to support our conclusion (please see answer to question #1 for clarification)

9. Also, loss-of-function of BIN2 should also be tested for effects on fertility in an EMS1-dependent pathway.

We agree that this experiment would enhance our understanding of BIN2 in EMS1-TPD1 signaling pathway, but BIN2 family has 10 members. To acquire the loss-of-function of BIN2 is currently impractical. But our main goal remains to relate the KNOWN BR signaling pathway to the unknown EMS1 signaling pathway with KNOWN BR signaling components. We believe that we have just done that.

10. Another important experiment is to show that BES1 and BZR1 are not dephosphorylated in the *ems1* mutant flowers, similar to the experiments shown for the *bri1* mutant.

In theory, this is a great experiment. However, *ems1* mutant has no tapeta where EMS1 presumably regulates BES1. Therefore, we cannot check the phosphorylation status of BES1 in tapeta when the tapeta do not exist, which means that this experiment is not doable. Analysis of dephosphorylation of BES1 in flowers will only detect BRI1-BR signaling but not EMS1-TPD1 signaling. Thus, it will not generate any new insight.

11. The experiments using homologs from *Physcomitrella* support the idea that these homologous proteins are similar to the Arabidopsis EMS1 and TPD1 protein to substitute EMS1 and TPD1 in Arabidopsis.

These experiments do not show that the *Physcomitrella* proteins can cause similar downstream changes in Arabidopsis; these proteins certainly do not support pollen development in Bryophytes because Bryophytes do not produce pollen.

Therefore, the experiments do not support the claim that EMS1 and TPD1 have conserved functions (biological or molecular) functions in land plants.

Because BRI1 can also rescue the *ems1* mutant when BRI1 is expressed from the EMS1 promoter, the experiments with PpEMS1 does not inform regarding the native function of PpEMS1.

First, we state that EMS1-TPD1 pair is conserved, meaning that they are found in all land plants that have completely sequenced genomes. Second, we state that EMS1-TPD1 might have a function (may be exapted later). Yet, this function can complement Arabidopsis mutants when there are expressed in them. However, we have not claimed that EMS1 and TPD1 have conserved functions (biological or molecular functions) in land plants, neither stated what the native function of PpEMS1 was. We are trying to point out that all land plants may have this signaling pair regardless how it works.

12. To investigate the conservation of EMS1 in land plants, experiments to test the function of

PpEMS1 in *Physcomitrella* must be performed, including expression studies and molecular genetic tests. Therefore, the conclusion stated in the subtitle on lines 277-288 is not supported by the results.

In this paper, we would like to focus on identifying the downstream components (at least one component) of EMS1 in *Arabidopsis*. We agree with the reviewer and have made changes in the text.

In summary, the manuscript contains largely confirmatory results and the results do not provide sufficient support for several conclusions. The overall manuscript does not lead to sufficient new insights regarding EMS1 function and related genes to warrant publication in *Nature Comm*. The manuscript in its current form is also not suitable in a more specialized journal, because the authors have not demonstrated a clearly understanding of the implications of the results, and have not presented the results properly.

We have taken these comments and suggestions very seriously. We almost rewrote the whole manuscript. We appreciate this reviewer for making our manuscript better.

Minor comments: there are numerous writing problems. The following are some examples.

1. Line 57, the sentence should be change to “apical meristem into the inflorescence and floral meristems by a combinative action of flowering transition and floral meristem genes”. In addition, the ref 7 is not a correct reference.

We have removed the whole statement.

2. Even for floral organ identity genes, ref 7 is for only one type of floral organ identity genes (E-function), but there are also other floral organ identity genes (ABC).

We have removed the whole statement.

3. Line 64, again, ref 7 is not a correct reference for floral meristem.

We have removed the whole statement.

4. Line 110, the sentence is missing a word at the end, such as “domain” or “segment”.

We have removed the whole statement.

5. Line 129, it is not clear what the authors meant by “common function”. Clearly these two genes affect different aspects of plant development. It is true that BRI1 and EMS1 are phylogenetically close compared with many other RLKs. However, many phylogenetically close genes have different biological functions. BRI1 and EMS1 also have distinct biological functions. We have removed the whole statement.

6. Line 131, the sentence is incomplete. BRI1 and EMS1?

I am sorry for the mistake and we have made the changes

7. Line 134, authors have indicated that BRI1 and EMS1 are known to share coreceptors, so it is not clear why they claim to have revealed this.

We agree with the reviewer and we have removed this claim.

8. There are other places where the statements about results or conclusions are inaccurate.

We have edited the whole text.

Reviewer #2 (Remarks to the Author):

The authors have addressed most of the concerns and I am convinced that the BRI1 and EMS1 pathways are sharing signaling components. I still think this work is of a high interest and it should be published in Nature Communications. I only have a few requests for a minor revision before publication:

- The co-expression of TPD1 and EMS1 partially complements *bri1-116* mutant phenotype. Actually, the plants show signs of increased levels of BR signaling based on the phosphorylation status of BES1, which is in state of constitutive ON state (Fig 2 d). Increased levels of activated BES1 can be explained by high levels of the ligand in the transgenic plants. The authors argue that TPD1 cannot be chemically synthesized and exogenously applied and they provide an elegant solution by crossing *proBRI1:EMS1* with three different *proBRI1:TPD1* transgenic lines with different TPD1 expression levels (FIG S4 a-b). However, the line (# 4) with higher TPD1 expression levels is phenotypically closer to wild type plants, while BES1 phosphorylation status is closer to wild type in line #12, which has the lowest TPD1 expression level. Additionally, the expression of CPD gene is slightly upregulated and BAS1 downregulated in transgenic EMS1&TPD1/*bri1-116* plants when compared to BRI1/*bri1-116* (Fig S5 c,e).

The author should indicate which transgenic lines were used for qPCR, #4, #15 or #12?

We used 6# × 15# for qPCR. We have noted it in the legend.

This result suggests that the BES1 activated by EMS1-TPD1 signaling pair, cannot perform exactly the same functions of BRI1-BR activated BES1 during maintenance BR homeostasis, since expression of these two genes shows opposite trend in increased BR signaling conditions. This might imply that BES1 transcription factor, although common for the two signaling pathways, is somehow differently controlled within the two pathways via perhaps, differential phosphorylation, interaction partners, etc? The authors should discuss this possibility in the discussion section?

We agree with the review, *EMS1 & TPD1/bri1-116* transgenic lines show some degree of variations in phenotypes, which was largely consistent with the accumulation of dephosphorylated BES1 in different transgenic lines. When the same transgenic line of *EMS1/bri1-116* was crossed to different transgenic lines of *TPD1/bri1-116* with different expression levels, the expression level of ligand TPD1 was largely consistent with the accumulation of dephosphorylation BES1. However, the dephosphorylation BES1 levels are nonlinear to the plant sizes. This indicates that the plants have ability to reset their growth. It is worth mentioning that *EMS1-TPD1/bri1-116* could not rescue to WT phenotypes. We attribute this phenotypic difference to their ligand difference in these two signaling systems. A line of evidence is that the chimeric receptor *BRI1-EMS1* completely restores the *bri1-116* phenotypes with almost identical BR response to that of WT. As a result, it is improper to directly compare *EMS1-TPD1/bri1-116* plants to WT plants for dephosphorylation BES1. Currently, the exact mechanism that controls EMS1-TPD1 signaling pathway through BES1 is still in its infancy. We are glad that the reviewer points out that there is a less increase in BAS1 in *EMS1-TPD1/bri1-116* plants to *BRI1/bri1-116* expression plants. We would argue that due to drastic ligand difference, it is surprised that they still regulate BR responsive genes (BR biosynthetic and catabolic genes) although TPD1 is not BRs. Therefore, these results support that the BES1 dephosphorylation by EMS1-TPD1 is rather similar to the BES1 dephosphorylation by BRI1-BRs. In fact, chimeric receptor BRI1-EMS1 completely resembles BRI1 in almost all aspects, further supporting that EMS1 functions similarly to BRI1. We have discussed this aspect in text as well.

- Authors included more information on the number of transgenic lines, stating that they used at least 20 independent transgenic lines. How many of those were analyzed per construct?

Although the phenotypic variations in *EMS1 & TPD1/bri1-116* and *PpEMS1 & PpTPD1/bri1-116* lines were discernable, the phenotypic variations in other transgenic lines of each construct was not. For the former case, we selected three lines that represent different statures for analysis. For the latter case, we chose a typical line for analysis.

- BES1 blots have been improved and have sufficient quality for publication.

Thanks for the comments

- The biggest issue for this manuscript is the quality of written English. The style of writing is still far from acceptable, in fact, has gotten even worse. As suggested during the first round of revision, authors need to send the text to an academic proofreader. Here are the examples of unsatisfactory language:

We now have sent out our text to an academic proofreader for final check-up.

Line 51, Avoid repeating terms within the sentence (cell surface in this case)

We have removed the whole statement.

Line 55, Plant life cycle starts with saprophytic seed germination, not plant life.

We have removed the whole statement.

Line 61, Do not start a sentence with "At the end". Better is Finally.

We have removed the whole statement.

Line 110, followed by transmembrane "domain".

We have removed the whole statement.

Line 122, "Activated BES1/BZR1 then negatively regulates BR biosynthesis but positively controls BR metabolism to control the hormonal homeostasis." Did authors mean catabolism instead of metabolism?

We agree with the reviewer and have made the changes

Line 146, "ligand-binding domains are evolved" – rephrase this.

We have removed the whole statement.

Line 148, EMS1 and BRI1 are not sister groups, they might belong to sister groups.

We have removed the whole statement.

Line 166, "mutants were indistinguishable by eyes" – rephrase. We have rewrote it

Line 167, "Furthermore, among all the reliable molecular markers, including sensitive to BRs and BR inhibitors, dephosphorylation of the transcriptional factor BES1/BZR1, downregulation of the expression of the BR biosynthetic genes and upregulation of the expression of the BR metabolic genes, used to judge the active BR response, only the BR metabolic gene BAS1 showed a lesser effect." This sentence is a typical example of poor writing style. Again, professional proofreading is necessary!

We rewrote it

Line 186, "Signaling pairs of EMS1-TPD1 and BRI1-BRs trigger similar functions." Title of this paragraph should be changed to Signaling pairs of EMS1-TPD1 and BRI1-BRs trigger similar molecular responses.

We have changed it

Line 193, erase together from this sentence

we have changed it

Line 224, "BES1 was accumulated" should stand BES1 is dephosphorylated and accumulates...

We have changed it

Line 262, " ... why do plants have both receptors?" Avoid layman's terms and fatalistic statements/questions.

We have changed it

Line 279, 327, 349, similar comment to line 262

We have changed them

Line 358, 391 Do not start sentence with Doing so.

We have changed it

Line 366, "Therefore, it will be best that EMS1 perceives TPD1 in tapeta while BRI1 binds BRs elsewhere to complementarily control the whole plant development through the same essential signaling pathway." Rephrase, there is no such a thing as a best solution for the plant.

We have rewritten it

Line 372, " The cell surface receptors perceive and transmit various signals from the outside of the cell and transmit into the inside of the cell to control a wide range of physiological and developmental processes." Another sentence that needs to be rephrased and written in a scientific manner.

We rewrote it

Line 404, "like nothing else ever known." Remove this statement.

We have removed it